

# Effective universality in quantum gravity

**Astrid Eichhorn[1], Peter Labus[2], Jan M. Pawlowski[1,3] and Manuel Reichert[1]⋆**

**1** Institut für Theoretische Physik, Universität Heidelberg,
Philosophenweg 16, 69120 Heidelberg, Germany
**2** International School for Advanced Studies, via Bonomea 265, 34136 Trieste, Italy
**3** ExtreMe Matter Institute EMMI, GSI Helmholtzzentrum für Schwerionenforschung mbH,
Planckstr. 1, 64291 Darmstadt, Germany

⋆ reichert@thphys.uni-heidelberg.de

## Abstract

**We investigate the asymptotic safety scenario for a scalar-gravity system. This system contains two avatars of the dynamical Newton coupling, a gravitational self-coupling and a scalar-graviton coupling. We uncover an effective universality for the dynamical Newton coupling on the quantum level: its momentum-dependent avatars are in remarkable quantitative agreement in the scaling regime of the UV fixed point. For the background Newton coupling, this effective universality is not present, but qualitative agreement remains.**


# 1 Introduction

In asymptotically safe quantum gravity the high-energy regime is governed by a non-Gaussian fixed point. This fixed point renders the ultraviolet (UV) behaviour finite, making the theory non-perturbatively renormalisable [1,2]. In recent years substantial evidence was collected in favour of this scenario [3–63], also in scalar-gravity systems [7–9,64–80]. See, e.g., [81–86] for reviews.

     The renormalisation group (RG)-idea behind asymptotic safety requires the split of the full metric $g_{\mu\nu}$ into a background metric $\bar{g}_{\mu\nu}$ and a dynamical fluctuation field $h_{\mu\nu}$ that carries the quantum fluctuations. Naturally, one of the most important ingredients in this approach is keeping track of diffeomorphism invariance and background independence. In the present work we address two questions that are linked to these key properties:

     The first question concerns the dynamical couplings of gravity-matter systems. Gauge theories feature different *avatars* of the gauge coupling, a prominent example being the running gauge couplings in the Standard Model. For instance, in QED the running electric coupling can be extracted from the wave function of the photon or from the electron-photon vertex. This relation follows from the Ward identities in QED. In QCD the running of the gauge coupling can be extracted from different combinations of vertex and propagator scalings including, e.g., the three-gluon vertex and the quark-gluon vertex. Again this can be derived from the identities following from the gauge symmetry, in this case the Slavnov-Taylor identities.

     In these examples the respective couplings are marginal and exhibit two-loop universality. This facilitates the identification. In gravity the above universality holds for the (marginal) $R^2$ and $R^2_{\mu\nu}$ couplings. However, the couplings in the classical Einstein-Hilbert action and the minimal couplings to matter are dimensionful and universality is not expected on the quantum level anymore. Still, the multi-graviton couplings related to Taylor expansions of the terms in the classical action, e.g. $1/G_N \sqrt{\det g}\, R$ and $1/G_N \sqrt{\det g}\, \Lambda$ agree on the classical level and are related by Slavnov-Taylor identities in quantum gravity. It is an intriguing physics question and of paramount technical importance, whether for all practical purposes these relations facilitate

an identification, for example, between all *avatars* of the minimal coupling in gravity, the Newton coupling $G_N$. We call this scenario *effective universality*, which is detailed in the next section. The quest for effective universality is motivated by its underlying physics properties. If realised it would hint at a near-perturbative nature of the asymptotically safe fixed point and, more importantly, at the physical nature of the fixed point as it is unlikely that a truncation artefact exhibits this property.

In the present work we investigate this question focusing on the dynamical pure gravity coupling and the dynamical gravity-scalar coupling. We stress that due to the dimensionful nature of the couplings, different *avatars* of the Newton coupling could agree if evaluated within the same scheme, but will of course depend, e.g., on the choice of regulator function in the context of an functional RG setup. In this work, universality is not to be understood in the sense of scheme-independence at the two-loop level.

We find that the fixed-point values and the leading coefficients of the above two couplings agree on a semi-quantitative level as a function of the number of minimally coupled scalars. These computations of dynamical couplings in a vertex expansion about a flat background extend the previous works of Refs. [3–10]. For extensions about a curved background, see, e.g. [11, 12].

The second question concerns the couplings of the background metric. These couplings are related via Ward-identities. Additionally, they are related to the dynamical couplings discussed above via Nielsen identities or split Ward identities. Importantly, these identities also carry the background independence of quantum gravity. In the present work we investigate to what extent the *avatars* of the background couplings can be identified with that of the fluctuation couplings. We emphasise that this identification is at the root of the background-field approximation whose background independence stands or falls with the validity of this identification.

In this work we do not only compare the avatars of the background Newton coupling to that of the fluctuation coupling, we also further improve the background coupling to a level-one coupling with the explicit use of a Nielsen identity. Related works on Nielsen identities in quantum gravity are [18–27]. We find that the effective universality that exists between the dynamical couplings is not present for the background and the level-one couplings. However, general qualitative features of the flow equations, such as as the sign of the scalar contribution, are preserved for all avatars of the Newton coupling.

## 2 Avatars of couplings and effective universality

In this section we explain the origin of different avatars of couplings in the effective action of matter-gravity systems. We further discuss their relation via the modified symmetry relations, STIs and Nielsen-identities, that are derived from the underlying diffeomorphism invariance and its breaking in the presence of cutoff terms. In short, effective universality is the notion that these complicated symmetry identities are well approximated by Ward identities, that is a diffeomorphism-invariant approximation of the effective action, for more details see subsection 2.3.

### 2.1 Avatars of couplings in matter-gravity systems

Asymptotically safe matter-gravity systems and their physics can be described in terms of the effective action $\Gamma[\bar{g}_{\mu\nu}, h_{\mu\nu}, \varphi]$ in a gauge-fixed setting. In the effective action we have dropped the Faddeev-Popov ghosts and restricted ourselves to the case with scalar matter described by $\varphi$. The field $h_{\mu\nu}$ denotes dynamical metric fluctuations around a generic background metric $\bar{g}_{\mu\nu}$. The occurrence of the latter comes hand in hand with the gauge fixing. The relation

between background metric and fluctuation field is not necessarily linear, but in the present work we consider the linear split

$$g_{\mu\nu} = \bar{g}_{\mu\nu} + \sqrt{Z_h G_N}\, h_{\mu\nu}, \tag{1}$$

with the graviton wave-function renormalisation $Z_h$ normalised to one at some RG scale $\Lambda$. In (1) we have dropped the wave-function renormalisations of $g_{\mu\nu}, \bar{g}_{\mu\nu}$ for the sake of readability. The $G_N$-factor leads to a fluctuation field $h_{\mu\nu}$ with the canonical dimension one. The effective action $\Gamma[\bar{g}_{\mu\nu}, h_{\mu\nu}, \varphi]$ is not diffeomorphism-invariant, but satisfies non-trivial Slavnov-Taylor identities (STIs). The information on physics is carried by the diffeomorphism-invariant effective action,

$$\Gamma[g_{\mu\nu}, \varphi] = \Gamma[g_{\mu\nu}, h_{\mu\nu} = 0, \varphi]. \tag{2}$$

and the respective background correlation functions are related to S-matrix elements. However, their computations requires the knowledge of the scattering processes with dynamical gravitons $h_{\mu\nu}$. These processes are described through different couplings $G_{\vec{n}}$, where the vector $\vec{n}$ consists of the numbers of the different dynamical fields $\Phi = (h_{\mu\nu}, c_\mu, \varphi, \dots)$ that take part in the process,

$$\vec{n} = (n_h, n_c, n_\varphi, \dots). \tag{3}$$

The couplings $G_{\vec{n}}$ of all these fields to the dynamical graviton are avatars of the gravitational self-coupling $G_N$. In the present study we concentrate on gravity-scalar couplings. This leaves us with couplings labelled by two indices only,

$$G_{(n_h, n_\varphi)}. \tag{4}$$

We denote dimensionless versions of the Newton coupling by capital letters, e.g. as above $G_{(n_h, n_\varphi)}$, and dimensionful versions with an additional over-bar, such as $\bar{G}_{(n_h, n_\varphi)}$. The fluctuation couplings $G_{(n_h, n_\varphi)}$ defined in (4) are related to the expansion coefficients in an expansion in powers of $h_{\mu\nu}$, the couplings $G_{(n_{\bar{g}}, n_\varphi)}$ are related to those in an expansion in powers of $\bar{g}_{\mu\nu}$. The respective vertices are given by

$$\Gamma^{(n,m,l)}(p_1, \dots, p_{n+m+l}) = \frac{\delta^{n+m+l}\Gamma[\bar{g}, h, \varphi]}{\delta\bar{g}^n(p_1, \dots)\delta h^m(\dots, p_i, \dots)\delta\varphi^l(\dots, p_{n+m+l})}, \tag{5}$$

where we suppress the indices on $\bar{g}$ and $h$ for brevity of notation. The couplings (4) are now defined by (5) at selected kinematic configurations. In this work, we focus on $G_{(3,0)}$ and $G_{(1,2)}$ defined at the momentum-symmetric point. The former coupling relates to the scattering of three gravitons, and is derived from the pure gravity part of the effective action. The coupling $G_{(1,2)}$ relates to the scattering of one graviton and two scalars, and is derived from the kinetic term of the scalars. This coupling is also present in the free -no self-interaction- scalar theory and can be considered the fundamental coupling of scalar fields to gravity. We compare these couplings based on their flow equations. We do not compute the explicit STIs that relate them. In general the information from the flow and the STIs is equivalent, but in a truncated non-perturbative computation they will not agree. It is an important task to quantify to what extent the STIs are satisfied but this goes beyond the scope of this work. The computation of the STIs is a technically challenging task, for an example in QCD see [87, 88].

Evidently these couplings cannot be defined uniquely and depend on the given kinematical limit. Note that this even holds for dimensionless couplings beyond one loop, despite their universal RG running. Accordingly, the evaluation of, e.g., scattering processes with different

momentum configurations requires an analysis of the corresponding $n$-point vertex as a function of all its independent momenta, i.e., a simple function of one momentum cannot capture the full dynamics adequately. If dealing with an approximation to the theory that does not maintain the full momentum-dependence of vertices, a typical choice is the symmetric point, for higher-order vertices *a symmetric point*. Using these momentum configurations can lead to semi-quantitative agreement with the full results even in strongly-correlated systems, for recent work in gravity see [6], and in QCD see [88]. For a related discussion in the effective field theory approach to gravity see [89]. Keeping this caveat in mind, we proceed with our study whether avatars of the Newton coupling, defined using the symmetric momentum configuration of various three-point vertices, show semi-quantitative agreement.

With the dynamical vertices (5) and the dynamical propagators we can compute the background vertices, that is the $S$-matrix elements. This leads to further avatars of the Newton coupling, this time being directly related to S-matrix elements for the selected momentum configuration. In the present work we consider the avatar of the Newton coupling of the background curvature term in the action. It is distinguished from the $G_{\vec{n}}$ by two properties: first it is the prefactor of a diffeomorphism invariant term in the action. Second, as a pure background quantity it does not drive the RG flow of the system, which is driven by the fluctuation field and its couplings. In this work, we refer to its dimensionless version as $\bar{G}$ and the dimensionful version as $G_N$.

## 2.2 RG-approach to asymptotically safe matter-gravity systems

The standard approach to computations in asymptotically safe gravity is the functional renormalisation group (FRG). The FRG approach to quantum gravity is based on the flow equation for the effective action, the Wetterich equation [90–92],

$$\partial_t \Gamma_k = \frac{1}{2} \text{Tr} \left[ G_k \, \partial_t R_k \right] . \tag{6a}$$

In (6a) we have introduced the RG-time $t = \log k/k_0$ with a reference scale $k_0$. The trace sums/integrates over the discrete/continuous spectrum of the propagator $G_k$. We emphasise that the flow of the effective action is solely driven by the second derivatives of the effective action w.r.t. the fluctuation fields. Accordingly it depends on the fluctuation propagators

$$G_k = \frac{1}{\Gamma_k^{(0,2)} + R_k} , \tag{6b}$$

with the second fluctuation field derivatives defined in (5), and the background-metric-dependent graviton and scalar field regulators $R_{h,k}$ and $R_{\varphi,k}$ respectively. While $\Gamma_k^{(0,2)}$ is a matrix in field space, which features off-diagonal components, the regulators are chosen to be diagonal. We have dropped the ghost contribution in (6a) for the sake of readability while taking it into account in our calculations. Induced scalar-ghost interactions [93] are neglected in our truncation.

Let us now come back to the question of the symmetry identities mentioned in subsection 2.1. While background diffeomorphism invariance is introduced as a mere computational tool and can even be established for theories without diffeomorphism invariance, it inherits the physical diffeomorphism invariance of gravity via the diffeomorphism STIs and the NIs/sWIs. The latter carry the background independence of gravity by relating derivatives with respect to the background metric $\bar{g}_{\mu\nu}$ and the fluctuation field $h_{\mu\nu}$.

In summary, the approach encodes the background independence and diffeomorphism invariance of observables in a counter-intuitive way: background independence of the setting implies nontrivial relations instead of simple equalities between couplings that would be

equal in a classical, diffeomorphism invariant setting. This also implies that diffeomorphism invariant approximations to $\Gamma[\bar{g}_{\mu\nu}, \Phi]$ are potentially at odds with physical diffeomorphism invariance and background independence. They should be taken with a grain of salt and have to be investigated thoroughly. The current work is a first step in this direction in a coupled matter-gravity system.

In the present renormalisation group setup the situation is even more intricate as diffeomorphism invariance and background independence are broken by the presence of the infrared regularisation. Any local coarse graining procedure requires the introduction of a background in order to define a notion of high-momentum modes. The presence of the corresponding cutoff term leads to *modified* STIs, NIs/sWI. In the limit $k \to \infty$ these deviations from the standard STI and sWI may play a crucial rôle for the correct description of the physical dynamics. In order to restore background independence in the physical limit $k \to 0$, the violation of diffeomorphism invariance and background independence introduced via the regulator must be compensated for by an appropriate UV initial condition. This UV initial condition violates diffeomorphism invariance and background independence such that the violation is fully 'eaten up' by the RG flow to the IR. Note also that the *physical* UV limit is the one where physical scales (momenta, curvature etc) take large values, but $k$ is kept at $k = 0$. Although physical scales can act as an IR cutoff, the UV limit $k \to \infty$ could show differences from the limit where physical scales such as momentum or curvature scales take their UV limit.

## 2.3 Effective universality

In the setup in subsection 2.1 and subsection 2.2 already one diffeomorphism-invariant operator at the classical level, for example the curvature scalar $\sqrt{g}R$ leads to infinitely many different couplings at the quantum level: These are obtained by taking the $n$th $h_{\mu\nu}$-derivative of $\sqrt{g}R$ and projecting $\Gamma^{(n)}$ (given a complete basis) on this tensor structure. While still being related by STIs their couplings do not agree. In the presence of the regularisation these STIs turn into mSTIs.

The situation is slightly different for the $n$th order background couplings: they even agree at the full quantum level as they are related by Ward identities due to background diffeomorphism invariance. This property even survives the introduction of the regularisation. However, as discussed above, the computation of their $\beta$-functions requires the knowledge of the fluctuation vertices. They are related to the background vertices by the Nielsen or split Ward identities, which turns the Ward identities into the STIs. In the presence of the regularisation we have modified NIs as we have mSTIs.

This leaves us with the technical challenge of computing all these coupling avatars related to a given operator, in the present example the avatars of the Newton coupling. Specifically, the challenge lies in the need to close a given system of flow equations for correlation functions that depend on the higher-order correlation functions. To that end one has to provide an ansatz for higher-order couplings for which the flow is not computed. The canonical choice is their classical value. In quantum gravity this canonical choice leads to an identification of all higher-order couplings derived from a given operator with the lowest-order one, effectively restoring diffeomorphism invariance. This we call *effective universality*.

For example, let us assume for a moment that we only compute the flow of one avatar of the Newton coupling. Then the canonical choice leads to the identification of all higher order Newton couplings with the lowest order one. If we apply this concept to the dynamical system, effective universality can be summarised by

$$G_{(n_h, n_\varphi)} \approx G, \qquad n_h, n_\varphi \in \mathbb{N}, \tag{7}$$

with a unique Newton coupling for a suitably chosen momentum configuration. One of the

main aims of this paper is to compare the scale dependence of these couplings under the impact of quantum fluctuations of the metric and of $N_s$ scalar fields.

In its maximal version for both, background couplings and fluctuation couplings, effective universality can be summarised in a concise form of the effective action,

$$\Gamma[\bar{g}_{\mu\nu}, h_{\mu\nu}, \varphi] = \Gamma_{\text{diff}}[\bar{g}_{\mu\nu} + h_{\mu\nu}, \varphi] + \Delta\Gamma_{\text{gauge}}[\bar{g}_{\mu\nu}, h_{\mu\nu}, \varphi], \tag{8}$$

with a diffeomorphism-invariant action $\Gamma_{\text{diff}}[g]$ and

$$\Delta\Gamma_{\text{gauge}}[\bar{g}_{\mu\nu}, h_{\mu\nu}, \varphi] \approx S_{\text{gf}}[\bar{g}_{\mu\nu}, h_{\mu\nu}] + S_{\text{gh}}[\bar{g}_{\mu\nu}, h_{\mu\nu}, c_\mu], \tag{9}$$

with gauge fixing and ghost action, $S_{\text{gf}}$ and $S_{\text{gh}}$, respectively, see [2, 94]. Furthermore, only the regulator terms would carry the breaking of background independence. This approximation is called the *background-field approximation*. It has been used predominantly in the RG approach to quantum gravity and is being paramount to effective field theory applications in quantum gravity [95–101].

In summary the quest for *effective universality* is directly related to the task of finding an efficient (rapidly convergent) expansion of the quantum effective action of matter-gravity systems in diffeomorphism-invariant operators. While this task is seemingly a technical one it is -in disguise- the quest for the aspects of physics that govern quantum gravity systems.

# 3 RG for scalar-gravity systems

In the present work we aim to shed light on the above issues and specifically explore in which settings effective universality may emerge in simple approximations. To that end we compare two avatars of the dynamical Newton coupling in this section. The first is defined from the three-graviton vertex, as in [5–7, 10], and its dimensionless version is called $G_{(3,0)}$. The second one is defined from the graviton-two-scalar vertex as in [8, 9], and is called $G_{(1,2)}$.

To project the RG flow (6a) onto a given coupling $G_{(n,m)}$ we take $n$ functional derivatives with respect to the graviton and $m$ with respect to the scalar field. The resulting tensor has $2n$ open indices that we contract with an appropriate tensor structure, see [5–7, 10] and [8, 9] for details. The resulting structure then depends on the momenta of the $n+m$ external legs of the respective couplings. For the two couplings mentioned above, we use a symmetric momentum configuration, where we set the angles between each pair of momenta to $2\pi/3$ and the magnitude of the momenta to $p$. The couplings in a vertex expansion are actually momentum-dependent functions. For the analytic results, we project at $p = 0$ to instead model them by a single number. For the numerical results, we instead utilise a bilocal projection at $p = 0$ and $p = k$ as in [5–7, 10]: This projection is motivated by the fact that the momentum integrals are peaked at $p \approx k$. Hence this momentum regime is more important for quantitative accuracy and partially also for capturing qualitative features of the flow of correlation functions. The bilocal projection partially captures the global momentum dependence. It has been tested successfully against the fully momentum-dependent results in [5–7, 10].

To define our truncation, we start from an Einstein-Hilbert action that is accompanied by a gauge fixing and ghost action, and a canonical kinetic term for the scalar fields,

$$S = -\frac{1}{16\pi G_N} \int d^4x \sqrt{g}\,(R - 2\Lambda) + S_{\text{gf}} + S_{\text{gh}} + \frac{1}{2}\sum_{i=1}^{N_s} \int d^4x \sqrt{g}\,g^{\mu\nu}\partial_\mu\varphi^i\partial_\nu\varphi^i. \tag{10}$$

We use the standard Faddeev-Popov gauge fixing procedure, with gauge fixing action

$$S_{\text{gf}} = \frac{1}{2\alpha} \int d^4x \sqrt{\bar{g}} \, F_\mu \bar{g}^{\mu\nu} F_\nu \,,$$

$$F_\mu = \bar{\nabla}^\nu h_{\mu\nu} - \frac{1+\beta}{4} \bar{\nabla}_\mu h^\nu{}_\nu \,. \tag{11}$$

We specialise the background metric to a flat Euclidean one, $\bar{g}_{\mu\nu} = \delta_{\mu\nu}$, and work with the values $\alpha = 0$ and $\beta = 1$ for the gauge parameters. This is a fixed point of the RG flow [102], as are all combinations $\alpha = 0$ and $\beta$. This choice of gauge parameters is technically favourable on a flat background since the poles of all modes of the classical graviton propagator coincide. Later in this work, for the level-one improvement, we also resort to the gauge choice $\alpha = \beta = 0$.

Next we insert the linear parameterisation (1) into the above action and subsequently expand the Einstein-Hilbert action up to fifth order and the kinetic term of the scalar up to the third order in the fluctuation field $h_{\mu\nu}$. Taking into account that all field monomials in the action evolve independently under the RG dynamics due to the breaking of diffeomorphism invariance, we introduce a separate dimensionful coupling for each vertex and denote it by $\bar{G}_{(n_h, n_\varphi)}$. We also distinguish between different avatars of the cosmological constant, introducing a dimensionful graviton mass parameter $\bar{\mu}$ associated to a mass-like term in the graviton propagator, and couplings $\bar{\lambda}_n$ associated to the momentum-independent part of the $n$-graviton vertex. In our approximation, the vertex functions are written as Einstein-Hilbert tensor structures with the appropriate substitutes of the cosmological and Newton constant

$$\Gamma_k^{(0,n,m)} = S^{(0,n,m)}(\mathbf{p}; \Lambda \to \Lambda_n, G_N \to \bar{G}_{(n,m)}) \,. \tag{12}$$

Here, $\mathbf{p} = (p_1, \ldots, p_{n+m})$ denotes the momenta of the external fields. Note that the pure gravity terms in (12) are proportional to $\bar{G}_{(n,0)}^{n/2-1}$ while the gravity-matter terms are proportional to $\bar{G}_{(n,m>0)}^{n/2}$. Furthermore, (12) is proportional to $Z_h^{n/2} Z_\varphi^{m/2}$ due to the rescaling of the fluctuation fields, see (1). This construction together with the choice of regulator assures that the wave-function renormalisations only enter via the corresponding anomalous dimensions $\eta_i$. These are defined via

$$\eta_i(p^2) := -\partial_t \ln Z_i(p^2) \,. \tag{13}$$

Schematically the scale dependent action reads

$$\begin{aligned}
\Gamma_k[\bar{g}, h, \varphi] = {}& \Gamma_k^{(0,0,0)}[\bar{g}] + \Gamma_k^{(0,1,0)}[\bar{g}]h \\
& + \frac{1}{2}\Gamma_k^{(0,2,0)}[\bar{g}]h^2 + \frac{1}{3!}\Gamma_k^{(0,3,0)}[\bar{g}]h^3 \\
& + \frac{1}{2}\Gamma_k^{(0,0,2)}[\bar{g}]\varphi^2 + \frac{1}{2}\Gamma_k^{(0,1,2)}[\bar{g}]h\varphi^2 + \ldots \,,
\end{aligned} \tag{14}$$

where we suppress contributions coming from ghost fields to improve readability. The ghost two-point function and the ghost contributions to the running of other $n$-point functions are taken into account in our work.

In truncations one challenge is to find a way to consistently close the infinite tower of flow equations for the vertices. Such a consistent closure is expected to lead to an enhanced robustness of the truncation. Specifically the flow of an $n$-point vertex depends on the $n + 2$-point vertices. We explicitly evaluate the flow of couplings with $n \leq 3$ and equate the higher-order couplings $G_{(4,0)}$, $G_{(5,0)}$, $G_{(2,2)}$ and $G_{(3,2)}$ as well as $\lambda_4$ and $\lambda_5$ with the lower-order couplings.

In summary we compute and evaluate the coupled flow equations of the scale dependent dimensionless quantities

$$\bar{\lambda},\ \bar{G},\ \mu,\ \lambda_3,\ G_{(3,0)},\ G_{(1,2)},\ \eta_h(p^2),\ \eta_\varphi(p^2),\ \eta_c(p^2). \tag{15}$$

The background couplings $\bar{\lambda}$ and $\bar{G}$ do not enter the flow and thus do not affect the fluctuation couplings. Their flow equations are analytic and derived using the York-decomposition [103, 104] with field redefinitions [28, 64]. The explicit pure gravity flow equation for our gauge is displayed in [6, 51]. The $N_s$-dependent part is gauge independent and thus equal to, e.g. [70]. The flow equations for $\mu$ and $\lambda_3$ are also analytic and agree with [7] (the coupling $G_{(1,2)}$ has to be disentangled from $G_{(3,0)}$ in the appropriate terms). The momentum dependence of the Newton couplings, $G_{(3,0)}$ and $G_{(1,2)}$, and the anomalous dimensions, $\eta_h$ and $\eta_\varphi$, is of importance and thus, following the discussion in [5–7, 10], it is preferable to evaluate these at finite momentum, which does not allow for analytic equations. Nevertheless the analytic version of these flows leads to qualitatively reliable results. The analytic and momentum-dependent versions of $G_{(3,0)}$, $\eta_h$, and $\eta_\varphi$ agree with [7]. Again $G_{(1,2)}$ has to be distinguished from $G_{(3,0)}$. The analytic version of $G_{(1,2)}$ is the same as in [9] while the momentum-dependent version is derived for the first time in this work.

The bilocal projection prescriptions for the Newton couplings lead to

$$\beta_{G_{(3,0)}} = \left(2 + 3\eta_h(k^2)\right)G_{(3,0)} - \frac{24}{19}\left(\eta_h(k^2) - \eta_h(0)\right)\lambda_3\, G_{(3,0)}$$

$$+ (32\pi)^2 \frac{64}{171}\, G_{(3,0)}^{1/2} \times \left(\text{Flow}_{\text{tt},G_{(3,0)}}^{(hhh)}(k^2) - \text{Flow}_{\text{tt},G_{(3,0)}}^{(hhh)}(0)\right),$$

$$\beta_{G_{(1,2)}} = \left(2 + \eta_h(k^2) + 2\eta_\varphi(k^2)\right)G_{(1,2)} + \frac{8}{3}\, G_{(1,2)}^{1/2}\, \text{Flow}_{\text{tt},G_{(1,2)}}^{(h\varphi\varphi)}(k^2). \tag{16}$$

Here, the notation tt, $G_{(3,0)}/G_{(1,2)}$ indicates a contraction of the right-hand side of the Wetterich equation with the projection operator onto the corresponding coupling as in [6, 7]. By $\text{Flow}_{\text{tt},G_{(3,0)}}^{(hhh)}(k^2)$ we refer to the right-hand side of the Wetterich equation, projected on three external $h_{\mu\nu}$ legs, each contracted with a transverse projector, and evaluated at the external momentum set to $p^2 = k^2$. The prefactors such as $\frac{24}{19}$ or $(32\pi)^2 \frac{64}{171}$ arise from the contraction of the respective tensor structures with the projection operators. For $\beta_{G_{(3,0)}}$ they are identical to [5–7] and for $\beta_{G_{(1,2)}}$ to [8] up to a rescaling of $h_{\mu\nu}$. The argument of $\text{Flow}(x)$ denotes the magnitude of the external momenta on the right-hand side, which are set to the momentum-symmetric point. Thus $\text{Flow}_{\text{tt},G_{(3,0)}}^{(hhh)}(0)$ is the right-hand side of the Wetterich equation projected onto contributions with three external gravitons and evaluated at vanishing external momentum. Due to the shift symmetry in the scalar sector, $\text{Flow}_{\text{tt},G_{(1,2)}}^{(h\varphi\varphi)}(0) = 0$. Note further that we compute the momentum-dependent anomalous dimensions with the approximation that we evaluate the anomalous dimension at $p^2 = k^2$ if it appears in an integral, see [7] for a discussion of this approximation.

For the derivation of e.g. the Flow-expressions in (16) we used the symbolic manipulation system *FORM* [105, 106] as well as the *FormTracer* [107] to trace diagrams.

## 4  Effective universality for the dynamical couplings

To address our first key question, we compare the $\beta$-functions and fixed-point results for the dynamical system including $G_{(3,0)}$, $G_{(1,2)}$, $\mu$ and $\lambda_3$. The $\lambda_n$ and in particular $\mu = -2\lambda_2$ play a special rôle due to the convexity of the effective action. To see this consider the effective action

for classical gravity. It is the double Legendre transform of the classical action. Accordingly, for positive cosmological constant it only agrees with the classical action for large enough curvature. Thus, even for a diffeomorphism-invariant action, the $\lambda_n$ are not necessarily the same. In summary, in the reduced system under investigation effective universality may only hold directly for $G_{(3,0)}$ and $G_{(1,2)}$ even if it is fully present. This leaves us with the two avatars of the Newton coupling while $\mu$ and $\lambda_3$ should be evaluated in dependence of $G_{(3,0)}$ and $G_{(1,2)}$ on a given trajectory.

Effective universality is necessarily broken at a finite cutoff scale as the regulators break diffeomorphism invariance. Accordingly it cannot hold quantitatively for all cutoff scales. It may hold at $k \to 0$, and potentially at $k \to \infty$. While the former physical case is evident, the latter case deserves some explanation: in the physics limit at $k = 0$ and for momentum scales $p \gg M_{\text{Planck}}$ we are in the scaling regime of the UV fixed point. If effective universality holds for $k = 0$, we have in particular $G_{(3,0)}(p^2) \approx G_{(1,2)}(p^2)$. If we now increase the cutoff scale, the scaling couplings are only changed for $p^2 \approx k^2$, i.e. the flows are local in momenta. Accordingly, the high-momentum behaviour of $G_{(i,j)}(p^2)$ can only be changed at large $k$, where one already probes the scaling regime. Then, self-similarity in the scaling regime entails that $k$-scaling and $p$-scaling agree. Hence, *physical* effective universality at $k = 0$ translates into effective universality of the cutoff-dependent couplings in the scaling regime.

We shall see that $G_{(3,0)}$ and $G_{(1,2)}$ indeed feature a semi-quantitative effective universality on scaling trajectories close to the UV fixed point.

## 4.1 Effective universality at the fixed point

We solve the flow equations of the fully coupled fluctuation system, $G_{(3,0)}$, $G_{(1,2)}$, $\mu$ and $\lambda_3$, identifying all higher-order gravity couplings with $G_{(n \geq 3,0)} = G_{(3,0)}$ and $\lambda_{n \geq 3} = \lambda_3$, and all graviton-scalar couplings with $G_{(n,m \geq 2)} = G_{(1,2)}$. The $\beta$-functions of the Newton couplings are given schematically in (16).

The resulting fixed-point values are shown in Fig. 1. We observe that both Newton couplings have similar fixed-point values that increase with $N_s$. The couplings $\mu$ and $\lambda_3$ remain approximately constant as a function of $N_s$. This already shows a qualitative effective universality for $G_{(3,0)}$ and $G_{(1,2)}$ that supports the reliability of computations where this property is used. The similar behaviour of the two avatars of the Newton coupling for all $N_s$ and the $N_s$-independence of $\mu$ and $\lambda_3$ suggests to first perform a detailed analysis at a fixed $N_s$ and then a subsequent one of the $N_s$-dependence. For the first part of the analysis we choose $N_s = 0$. This is in complete analogy to the quenched approximation in QCD, where one drops all closed quark loops. In the present case it amounts to dropping all closed scalar loops. This does not imply the complete absence of scalar fluctuations, as they still appear in diagrams with internal scalar and graviton lines, i.e. in the diagrams of the $G_{(1,2)}$ flow.

## 4.2 Effective universality for quenched quantum gravity

In the quenched limit, $N_s = 0$, a quantitative self-consistency analysis reveals an even more interesting property than the mere similarity observed in Fig. 1. To that end we remind ourselves that the bilocal projection used in the present fixed-point computation is based on observations in the pure gravity system in [5, 6] for $G_{(3,0)}$ and $G_{(4,0)}$. In particular in [6] it was shown that the momentum dependence of the coupling $G_{(3,0)}$ and $G_{(4,0)}$ related to the curvature term $R$ is quantitatively given by a linear $p^2$-dependence. The four-graviton vertex has an additional $p^4$-dependence related to the $R^2$-term, no higher-order momentum-dependence is present, for more details see [6]. These properties are based on non-trivial cancellations between diagrams based on diffeomorphism invariance. This situation suggests the following self-consistency analysis of effective universality for $G_{(3,0)}$ and $G_{(1,2)}$: assume that effective

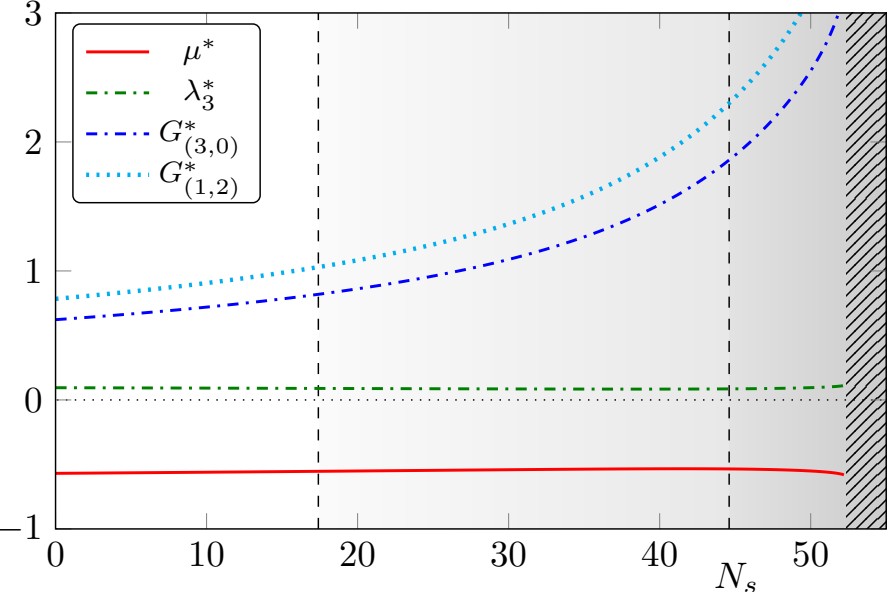

Figure 1: Fixed-point values for the fluctuation couplings as a function of $N_s$. The vertical lines at $N_s \approx 17.5$ and $N_s \approx 44.6$ show where $\eta_h(0)$ and $\eta_h(k^2)$ exceed two, respectively.

universality works quantitatively at the fixed point, that is

$$G_{\vec{n}} = G \qquad \text{with} \qquad G = G_{(3,0)}, \tag{17}$$

for all avatars of the Newton coupling. The self-consistency of (17) is tested quantitatively by evaluating the momentum-dependent $\beta$-functions $\beta_{G_{(3,0)}}$ and $\beta_{G_{(1,2)}}$ on (17) and the approximately $N_s$-independent fixed-point values $\mu^*$ and $\lambda_3^*$. Solving the momentum-dependent $\beta$-functions, cf. (16), for the momentum-dependent couplings on the bilocal fixed-point values leads us to

$$p^2 \sqrt{G_{(3,0)}(p^2)} \simeq -\frac{64}{171}(32\pi)^2 \left. \frac{\text{Flow}^{(3,0)}(p^2) - \text{Flow}^{(3,0)}(0)}{2 + 3\eta_h(p^2)} \right|_{G^*,\mu^*,\lambda_3^*},$$

$$p^2 \sqrt{G_{(1,2)}(p^2)} \simeq -\frac{8}{3} \left. \frac{\text{Flow}^{(1,2)}(p^2)}{2 + \eta_h(p^2) + 2\eta_\varphi(p^2)} \right|_{G^*,\mu^*,\lambda_3^*}. \tag{18}$$

Note that $\eta_\varphi(p^2) = 0$ in the chosen gauge, $\beta = 1$ and $\alpha = 0$. The momentum-dependent fixed-point couplings are shown in Fig. 2. There we have shifted $p^2 \sqrt{G_{(1,2)}(p^2)}$ by a constant in order to make the quantitatively coinciding linear dependence for $p^2 \gtrsim 0.3\,k^2$ apparent, see (20) evaluated at $p^2 = k^2$ for the definition of the shift. This coincidence is a non-trivial consequence of the different contributions of $\mu$ and $\lambda_3$ to both $\beta$-functions. It entails effective universality on the quantitative level. The deviation from effective universality at small momenta may have two different sources: first we expect that the regulator-induced breaking of effective universality is maximal at low momenta in comparison to the cutoff scale. A second source of the deviation may be the graviton mass scale in the graviton propagators, and could be related to the convexity-enforcement at work in the effective action.

Typically the momentum dependence of $p^2 \sqrt{G_{(1,2)}(p^2)}$ is studied in terms of a derivative expansion about $p^2 = 0$. For our results this corresponds to a small $p^2$-term and large $p^4$ and $p^6$-terms. This suggests an interpretation of the present results as the dominant generation

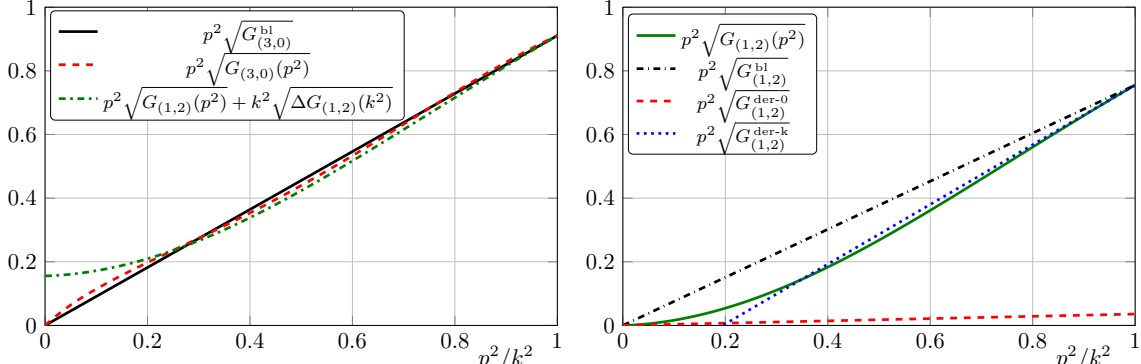

Figure 2: Displayed are the momentum-dependent Newton couplings evaluated on the quenched bilocal fixed point using effective universality for the momentum-independent couplings, see (17). Left: $p^2\sqrt{G_{(3,0)}(p^2)}$ and $p^2\sqrt{G_{(1,2)}(p^2)} - k^2\sqrt{\Delta G_{(1,2)}(k^2)}$, see (19). The black lines guides the eye and corresponds to the bilocal approximation $p^2\sqrt{G^{\mathrm{bl}}_{(3,0)}}$. Right: $p^2\sqrt{G_{(1,2)}(p^2)}$ as well as local and bilocal approximations to the momentum dependence of it.

of higher-order couplings such as $\sqrt{g}R_{\mu\nu}\varphi\nabla_\mu\nabla_\nu\varphi$. On the other hand, an expansion about a momentum $p^2 \gtrsim 0.3\,k^2$ leads to a $p^2$-coefficient of the same order as in the three-graviton vertex, which supports the emergence of effective universality. At the same time the higher-order momentum coefficients of the expansion $p^2 \gtrsim 0.3\,k^2$ are typically smaller by a factor three in comparison to the leading one. The expansion about $p = 0$ and its interpretation as higher-order operators hinges on a diffeomorphism-invariant expansion of the effective action deep in the regime where the regulator spoils diffeomorphism-invariance. Interestingly, the momentum dependence of $p^2\sqrt{G_{(3,0)}(p^2)}$ has a much clearer interpretation: the $p^2$-term is rather independent of the projection scheme.

This discussion and the highly non-trivial result of quantitative effective universality in Fig. 2 suggests to take a closer look at how well local and bilocal approximations capture the full momentum dependence and how they accommodate effective universality. Accordingly, we close this section on quenched quantum gravity with a discussion and evaluation of different approximation schemes.

### 4.2.1 Quantitative bilocal schemes

The full momentum dependences of the Newton couplings are well approximated by using the bilocal result for the three-graviton coupling, $G^{\mathrm{bl}}_{(3,0)}$, and by $G^{\mathrm{bl}}_{(3,0)}$ with an additional interpolating piece for the scalar-graviton coupling. This amounts to

$$p^2\sqrt{G^{\mathrm{quant}}_{(1,2)}(p^2)} := p^2\sqrt{G^{\mathrm{bl}}_{(3,0)}} + k^2\sqrt{\Delta G_{(1,2)}(p^2)}, \tag{19a}$$

with

$$\sqrt{\Delta G_{(1,2)}(p^2 \gtrsim 0.3\,k^2)} = \sqrt{G_{(1,2)}(k^2)} - \sqrt{G^{\mathrm{bl}}_{(3,0)}},$$

$$\Delta G_{(1,2)}(0) = 0, \tag{19b}$$

and $\Delta G_{(1,2)}(p^2)$ interpolates between these two values in the interval $0 \leq p^2 \lesssim 0.3\,k^2$, see Fig. 2. The accurate determination of this interpolation at small momenta $p^2 \lesssim 0.3\,k^2$ is numerically irrelevant as these momenta are suppressed in loops due to the $p^3$-factor from the integration measure.

We can make maximal use of the numerical irrelevance of the low-momentum regime with $p^2 \lesssim 0.3\,k^2$ and drop the non-linear piece altogether. This amounts to

$$\sqrt{\Delta G_{(1,2)}(p^2)} = \sqrt{G_{(1,2)}(k^2)} - \sqrt{G_{(3,0)}^{\text{bl}}}, \tag{20}$$

see also Fig. 2. In this approximation of the vertex $p^2 \sqrt{G_{(1,2)}(p^2)}$ does not vanish at $p^2 = 0$, which breaks shift symmetry. However, the approximation scheme never uses this information effectively restoring shift symmetry.

### 4.2.2 Qualitative bilocal schemes

A simpler approximation is dropping $\Delta G_{(1,2)}$ completely, $\Delta G_{(1,2)} \equiv 0$. With (19a) this leads to

$$G_{(1,2)}^{\text{qual}} = G_{(3,0)}^{\text{bl}}. \tag{21}$$

This leads to explicit shift symmetry in (19) but also triggers up to a $\sim 20\%$ deviation in the results of the respective diagrams proportional to $G_{(1,2)}$. This approximation has been used with $\Delta G_{(n,m)} = 0$ for all $n, m$ in (19) in matter-gravity systems in [7, 10, 108], and the results there receive now support by effective universality in the scalar-gravity system.

The final variant of the bilocal scheme is the standard bilocal approximation for $G_{(1,2)}$. Using shift symmetry with $p^2 \sqrt{G_{(1,2)}(p^2)}\big|_{p^2=0} = 0$, we are led to

$$G_{(1,2)}^{\text{bl}} = G_{(1,2)}(k^2), \tag{22}$$

for the respective coupling see the right panel of Fig. 2. It is up to $\sim 20\%$ bigger than $G_{(1,2)}(p^2)$ in the numerically relevant regime with $p^2 \gtrsim 0.3\,k^2$. Accordingly, it has a quantitative error of about this size but maintains explicit shift symmetry. Note also that it is $\sim 20\%$ smaller than the *slope* of $p^2 \sqrt{G_{(1,2)}(p^2)}$ for $p^2 \gtrsim 0.3\,k^2$ where effective universality takes place. This is the approximation we used for the fixed-point results in Fig. 1 and also use later in subsection 4.4. Based on these observations we call a deviation from effective universality of up to 20% a semi-quantitative agreement of the $\beta$-functions.

### 4.2.3 Derivative expansions

A derivative expansion is a local expansion in momenta. The expansion point is either chosen for analytic and numerical convenience or in order to optimise the convergence to the full result. Analytic convenience singles out $p^2 = 0$ as this allows for analytic flow equations for specific regulators such as the Litim regulator [109, 110] or the sharp cutoff.

Good convergence is usually achieved with an expansion about the momentum value where the integrands in the flow peak. This typically is a momentum close to the cutoff scale, $p^2 \approx k^2$, leading to

$$\sqrt{G_{(1,2)}^{\text{der-k}}} = \sqrt{G_{(1,2)}(k^2)} + p^2 \frac{\partial \sqrt{G_{(1,2)}(p^2)}}{\partial p^2}\Bigg|_{p^2=k^2}. \tag{23}$$

In the present case this has the additional benefit that it also includes a good estimate of the linear piece of the $G_{(1,2)}$ avatar of the Newton coupling, see the right panel of Fig. 2. Qualitatively is is in the same ballpark as the quantitative bilocal approximation with (20) described in subsubsection 4.2.1.

It is left to discuss the standard derivative expansion with the expansion point $p^2 = 0$ with

$$G_{(1,2)}^{\text{der-0}} = G_{(1,2)}(0). \tag{24}$$

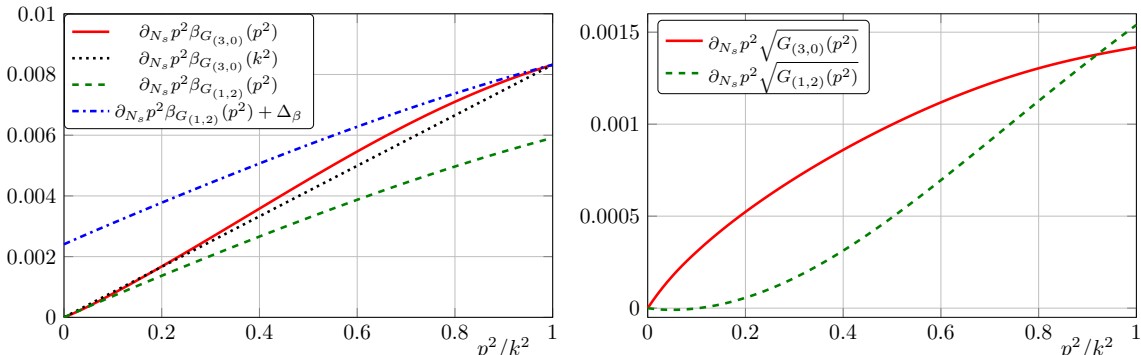

Figure 3: $N_s$-derivative of the momentum-dependent $\beta$-functions (left) and Newton couplings (right) evaluated on the bilocal fixed point for $N_s = 0$.

We note that the non-trivial momentum dependence of the vertex $p^2\sqrt{G_{(1,2)}}$ at small momenta $p^2 \to 0$ casts some doubt on the naive use of derivative expansions in quantum gravity. Moreover, the analysis of the momentum-space expansion about $p^2 = 0$ also applies to curvature expansions as used in the background-field approximation. Hence the current reliability discussion translates to computations within the background-field approximation.

Clearly, approximating $G_{(1,2)}$ by the derivative of $p^2\sqrt{G_{(1,2)}(p^2)}$ at $p^2 = 0$ leads to a significant deviation (factor $\sim 20$) of the resulting coupling from the full result in the numerically relevant regime for $p^2 \gtrsim 0.3\,k^2$. This issue has been already discussed in [5,6] for pure quantum gravity, where the deviation is smaller. This is already visible from the full momentum-dependence of $G_{(3,0)}$ in Fig. 2. Still, this scheme captures all qualitative aspects of the current system.

### 4.3 Effective universality for unquenched quantum gravity

The quantitative self-consistency analysis in subsection 4.2 above was done at $N_s = 0$. Now we study the $N_s$-derivative at $N_s = 0$. This gives us a sum of the different terms that show up with a linear $N_s$-dependence that comes from closed scalar loops. If this sum again shows a behaviour as seen in Fig. 2, this indicates the persistence of quantitative effective universality for all $N_s$. An analysis for large $N_s$ is hampered by the strongly rising Newton couplings: we hit the reliability bounds of the approximation before the $N_s$-effects become dominant. Thus our analysis is sufficient at the current level of the approximation.

We first examine the $N_s$-derivative of the $\beta$-functions of the Newton couplings evaluated at the fixed point and then study the $N_s$-dependence of the momentum-dependent Newton couplings. For the $N_s$-derivative of the $\beta$-functions we only take the explicit $N_s$-dependence into account, whereas the $N_s$-dependence of the fixed-point values is neglected. They are given by

$$\partial_{N_s} p^2 \beta_{G_{(3,0)}}(p^2) \simeq 3\,G\,p^2\,\partial_{N_s}\eta_h(p^2)$$

$$+ (32\pi)^2\,\frac{64}{171}\sqrt{G}\,\partial_{N_s}\left[\text{Flow}^{(3,0)}(p^2) - \text{Flow}^{(3,0)}(0)\right],$$

$$\partial_{N_s} p^2 \beta_{G_{(1,2)}}(p^2) \simeq G\,p^2\,\partial_{N_s}\eta_h(p^2) + \frac{8}{3}\sqrt{G}\,\partial_{N_s}\text{Flow}^{(1,2)}(p^2).\tag{25}$$

This is displayed in the left panel of Fig. 3. We see a qualitatively similar momentum-dependence of both, $\partial_{N_s}\beta_{G_{(3,0)}}(p^2)$ and $\partial_{N_s}\beta_{G_{(1,2)}}(p^2)$ evaluated at the fixed point. Interestingly, the momentum slope of the two $\beta$-functions is almost identical at $p^2 = 0$ and at $p^2 = k^2$. Moreover, their

absolute values are one/two orders of magnitude smaller than the quenched $\beta$-functions, see Fig. 2. Accordingly, scalars only change the system qualitatively for $N_s \gtrsim 10 - 10^2$. This is seen in Fig. 1: the system is basically unchanged for $N_s \lesssim 20$. From there on the approximation violates reliability bounds for the regulator with $\eta_h \leq 2$, see [7], and should be taken with a grain of salt. In general we observe that the $\beta$-functions increase with increasing number of scalars, and hence also the fixed-point values increase with $N_s$.

The $N_s$-derivative of the fixed-point dressing of the vertices for the three-graviton coupling read

$$\frac{\partial_{N_s} p^2 \sqrt{G_{(3,0)}(p^2)}}{p^2 \sqrt{G_{(3,0)}(p^2)}} = \frac{\partial_{N_s}\left[\text{Flow}^{(3,0)}(p^2) - \text{Flow}^{(3,0)}(0)\right]}{\text{Flow}^{(3,0)}(p^2) - \text{Flow}^{(3,0)}(0)} - 3\frac{\partial_{N_s}\eta_h(p^2)}{2 + 3\eta_h(p^2)}, \tag{26a}$$

and for the minimal scalar-graviton coupling

$$\frac{\partial_{N_s} p^2 \sqrt{G_{(1,2)}(p^2)}}{p^2 \sqrt{G_{(1,2)}(p^2)}} = \frac{\partial_{N_s}\text{Flow}^{(1,2)}(p^2)}{\text{Flow}^{(1,2)}(p^2)} - \frac{\partial_{N_s}\eta_h(p^2)}{2 + \eta_h(p^2)}, \tag{26b}$$

where the second terms on the right-hand sides of (26) take care of the $Z^{n/2}$ dressing of the $n$-point vertices. The terms in the respective flow contributions from $\partial_{N_s}\text{Flow}^{(n,m)}$ read

$$\partial_{N_s}\text{Flow} = \left.\frac{\partial}{\partial N_s}\right|_{G,\eta_h}\text{Flow} + \left[\partial_{N_s}\eta_h(k^2)\right]\partial_{\eta_h}\text{Flow} + \frac{3}{2}\frac{\partial_{N_s}G}{G}\text{Flow}. \tag{27}$$

The first term on the right-hand side of (27) simply counts the number of scalars in closed scalar loops, which rises linearly with $N_s$. This term vanishes for $G_{(1,2)}$ as its flow has no diagram with a closed scalar loop. The second term takes into account the $N_s$-dependence of the graviton propagator as well as that of the wave-function renormalisations in the vertices. With the present RG-adjusted graviton regulator that is proportional to $Z_h$, this dependence is stored solely in the $\eta_h$-dependence of the scale derivative of the regulator. The anomalous dimension $\eta_h$ has a linear $N_s$-dependence proportional to the closed scalar loop for the graviton propagator. Together with the closed scalar loop for $G_{(3,0)}$ it gives the $N_s$-dependence at one-loop. For universal couplings such as the gauge couplings in the Standard Model these terms provide the universal $N_s$-dependence of the couplings.

The additional terms account for the typical resummations present in FRG computations: an additional contribution in the $\partial_{N_s}\eta_h$-derivative takes into account the $N_s$-dependence of the fixed-point coupling. The third term takes into account the $N_s$-dependence of the fixed-point coupling from the prefactor $G^{3/2}$ in all the diagrams. Further terms are present that take into account the $N_s$-dependence of $\mu$ and $\lambda_3$, which are dropped in the present analysis as they are approximately $N_s$-independent at the fixed point, see Fig. 1.

This leads us to the right panel of Fig. 3, which displays the $N_s$-derivative of the fixed-point couplings $p^2 \sqrt{G_{(3,0)}(p^2)}$ and $p^2 \sqrt{G_{(1,2)}(p^2)}$. The momentum dependence encodes an intriguing structure. First of all the quantitative effective universality present in Fig. 2 is not found. Still, the $N_s$-dependences have the same size, which explains the similar growth in Fig. 1. Note that this similarity is even better for the momentum regime relevant in the loop integrals with $p^2 \approx k^2$, so fully momentum-dependent or bilocal approximations take account of this fact.

The momentum-dependence in the right panel of Fig. 3 is not reflected fully by a linear function in $p^2$. This suggests that higher-order terms are generated by the unquenching terms, i.e. the closed scalar loops. We emphasise that the momentum-dependence in the right panel of Fig. 3 is a superposition of Fig. 2 and the left panel of Fig. 3. In the latter figures the momentum slopes did agree well in the momentum regime $p^2 \sim k^2$. Consequently, also the absolute

difference in the momentum slopes at $p^2 \sim k^2$ is small in the right panel of Fig. 3. However, the relative difference in the momentum slope is large. Note that values of the resummed $N_s$-derivative of the momentum-dependent fixed-point vertices in the right panel of Fig. 3 are even an order of magnitude smaller than the $N_s$-derivative of the momentum-dependent $\beta$-functions in the left panel of Fig. 3. Thus one might potentially interpret our result as indicating that the $N_s$-dependence of the momentum-dependent fixed-point vertices is actually compatible with zero.

As in the quenched scalar-gravity system we now discuss the relation of the momentum-dependent vertex function to diffeomorphism-invariant operators by Taylor expanding about vanishing momentum. Again, this hinges on the existence of a diffeomorphism-invariant expansion in the deep UV, where the regulator is expected to spoil diffeomorphism-invariance. Nonetheless it is worthwhile to examine the overlap of operators with vertex functions and to check, which inclusion of operators might restore effective universality in the unquenched sector.

The $p^2$-terms in $G_{(3,0)}$ and $G_{(1,2)}$ stand for the curvature scalar $\sqrt{g}R$ and kinetic term $\sqrt{g}\varphi\Delta_g\varphi$, respectively. The $p^4$-terms in $G_{(3,0)}$ and $G_{(1,2)}$ relate to $\sqrt{g}R_{\mu\nu}^2$ and $\sqrt{g}R_{\mu\nu}\varphi\nabla_\mu\nabla_\nu\varphi$ terms respectively, see, e.g. [9, 56]. Note also that our projection has no overlap with $\sqrt{g}R^2$ and $\sqrt{g}R\varphi\Delta_g\varphi$. The $\sqrt{g}R^2$ term is generated with a coupling of the same order of magnitude as $G_{(3,0)}$ already in the Einstein-Hilbert truncation, see [6], and it is only our projection that has no overlap with it. In turn, the $R_{\mu\nu}^2$ coupling generated by the Einstein-Hilbert truncation is compatible with zero, see also Fig. 2. The present finding suggests that the unquenching effects due to the closed scalar loops generate corrections of the $G_{(3,0)}$-avatar of the Newton coupling as well as an $R_{\mu\nu}^2$ coupling of a comparable size. The generation of the latter is particularly intriguing as the respective term is not generated in quenched scalar-gravity systems. Consequently, it might be worthwhile to investigate effective universality in the unquenched scalar-gravity-system in the presence of $\sqrt{g}R_{\mu\nu}^2$ and $\sqrt{g}R_{\mu\nu}\varphi\nabla_\mu\nabla_\nu\varphi$ operators.

## 4.4 Effective universality beyond the fixed point

The intriguing result displayed in Fig. 2 has shown that effective universality holds quantitatively at the fixed point within the $N_s$-range of validity of the current approximation. As argued in the beginning of Sec. 4, we expect effective universality to only hold in the vicinity of the fixed point, that is on given trajectories for $k \to \infty$, and, if present, for all momenta at $k \to 0$.

Such a scenario suggests an approximation that utilises effective universality also for finite cutoffs as the related error disappears at $k = 0$ and $k \to \infty$. Here we investigate the question how it fares away from the fixed point. For the sake of simplicity we do not resort to the quantitative bilocal scheme described in subsubsection 4.2.1, but to the qualitative bilocal scheme described in subsubsection 4.2.2 with (22). If evaluating the $\beta$-functions on (17) and on the fixed point values of $\mu$ and $\lambda_3$ we obtain

$$\beta_{G_{(3,0)}}\Big|_{\mu^*,\lambda_3^*} = 2G - (3.4 - 0.013N_s)G^2 + \mathcal{O}(G^3),$$

$$\beta_{G_{(1,2)}}\Big|_{\mu^*,\lambda_3^*} = 2G - (2.7 - 0.0085N_s)G^2 + \mathcal{O}(G^3). \tag{28}$$

In (28) we have used the fixed-point values of $\mu$ and $\lambda_3$ at $N_s = 0$, as they are almost $N_s$-independent (cf. Fig. 1). The coefficients of the $\beta$-functions (28) do not feature an effective universality. This is caused by the missing offset in (19) at momenta $p^2 \gtrsim 0.3\,k^2$ required for the quantitative agreement, as explained in subsection 4.1.

As our further investigation of effective universality is based on the $\beta$-functions in (28) we confirm here that the qualitative nature of the present approximation scheme discussed in

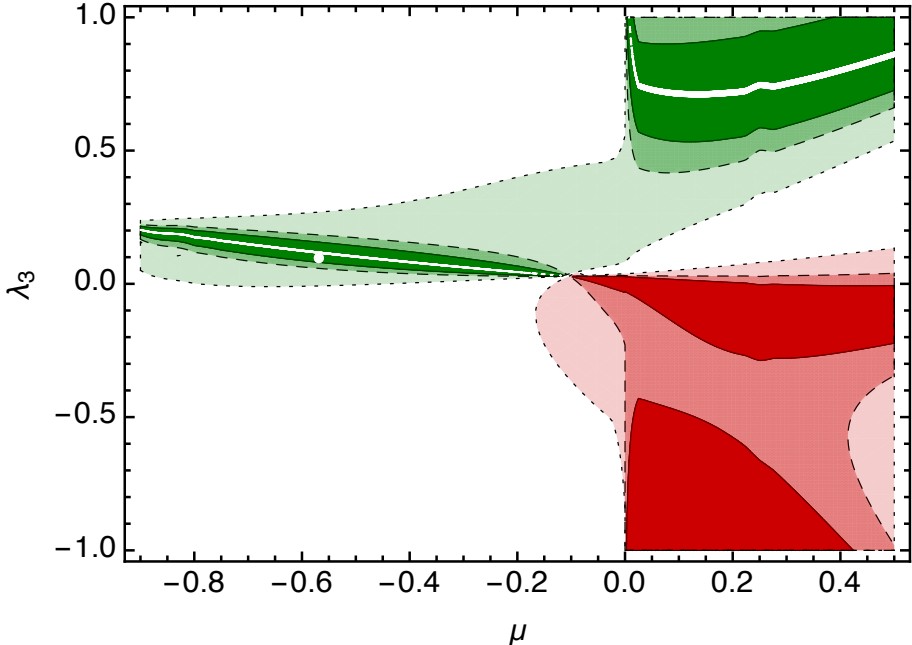

Figure 4: The regions in which $\varepsilon < \frac{1}{5}$ ($\varepsilon < \frac{1}{3}$; $\varepsilon < \frac{3}{4}$) are marked in dark (light, dashed contour; lighter, dotted contour) colours for $N_s = 1$ and $G \to 0$. The green (red) colour indicates $\Delta\beta_G < 0$ ($\Delta\beta_G > 0$). The two green regions are centred around a white line where $\varepsilon = 0$. The bilocal UV fixed point is indicated by a white dot. The area is almost $N_s$ independent, as are the fixed-point values of $\mu$ and $\lambda_3$ (cf. Fig. 1).

subsubsection 4.2.2 below (22) already explains all the deviations in (28): The $N_s$-independent terms should be subject to underestimating the *slope* of $p^2 G^*_{(1,2)}$ in the regime $p^2 \gtrsim 0.3 \, k^2$ where effective universality takes place, see Fig. 2. Accordingly, this part of $\beta_{G_{(1,2)}}$ should be $\sim 20\%$ smaller than that of $\beta_{G_{(3,0)}}$ and their ratio should be $\sim 0.8$. From (28) we obtain $2.7/3.4 = 0.79$.

In the $N_s$-dependent part we did not find full effective universality, cf. Fig. 3. However the coefficients of these terms are two orders of magnitudes smaller than the pure gravity terms and thus they do not affect the present discussion within the $N_s$-range of validity of the current approximation, i.e. for $N_s < 50$.

In summary, (28) fully reflects the quantitative universality in the given bilocal approximation precisely by its semi-quantitative or qualitative pattern. This has to be kept in mind if evaluating deviations from effective universality within this approximation.

Now we proceed with this evaluation by devising a measure of the breaking of effective universality. It is a property of the anomalous part $\Delta\beta$ of the $\beta$-function,

$$\Delta\beta_{G_{(i,j)}} = \beta_{G_{(i,j)}} - 2G. \tag{29}$$

As a measure of effective universality we use the relative error between the scaling of two avatars of the coupling evaluated under the assumption of universality, that is $G_{\vec{n}} = G$ for the avatars of the Newton coupling. In the present case this reads

$$\varepsilon(G, \mu, \lambda_3, N_s) = \left| \frac{\Delta\beta_{G_{(3,0)}} - \Delta\beta_{G_{(1,2)}}}{\Delta\beta_{G_{(3,0)}} + \Delta\beta_{G_{(1,2)}}} \right|_{G_{\vec{n}}=G}. \tag{30}$$

In the simplest case of effective universality $\varepsilon$ is zero. In the present non-trivial realisation we have a breaking pattern for small momenta, see Fig. 2. In the presence of such a breaking

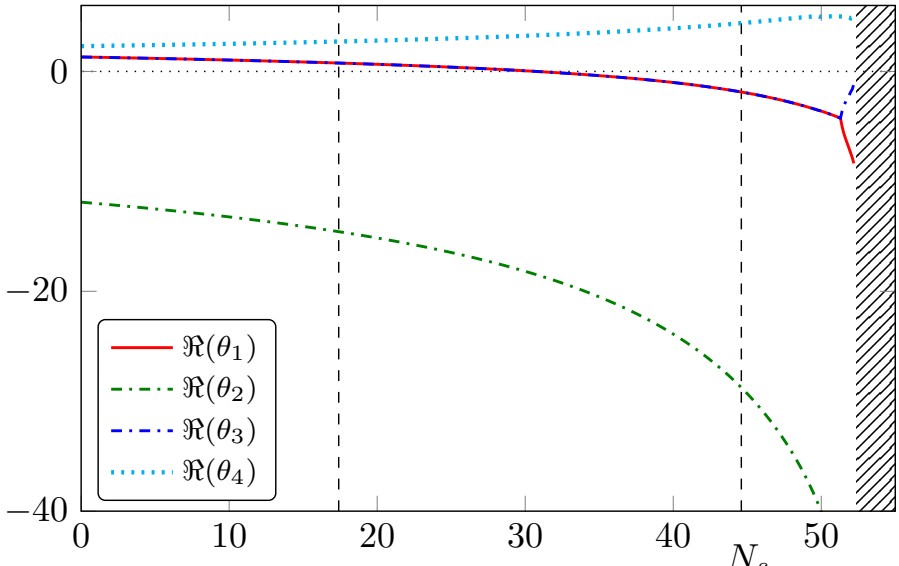

Figure 5: Critical exponents of the UV fixed point as a function of $N_s$, see Fig. 1 for the fixed-point values. The colours of the critical exponents indicate with which coupling the corresponding eigenvector has the largest overlap. The vertical lines at $N_s \approx 17.5$ and $N_s \approx 44.6$ show where $\eta_h(0)$ and $\eta_h(k^2)$ exceed two, respectively.

$\varepsilon = 0$ does indeed indicate a small violation of effective universality, while a small value might indicate its full quantitative presence.

Further we observe, that for $\varepsilon < 1$ the anomalous parts of the $\beta$-functions have the same sign, and different signs for $\varepsilon > 1$. In the limit $G \to 0$ we compare the $G^2$-terms as displayed in (28). The definition (30) also allows to separately compare the gravity and scalar contributions by taking the limits $N_s \to 0$ and $N_s \to \infty$, respectively. It does, however, not distinguish between anomalous parts of a $\beta$-function that allow for a UV fixed point at positive Newton coupling ($\Delta\beta < 0$) and that do not ($\Delta\beta > 0$).

In Fig. 4 we show the regions in the $(\mu, \lambda_3)$-plane where effective universality is realised for the coupling $G_{(3,0)}$ and $G_{(1,2)}$. In particular we display the regions in which $\varepsilon < \frac{1}{5}$ and $\varepsilon < \frac{1}{3}$ for $N_s = 1$ and $G \to 0$. We further distinguish between regions that allow for a UV fixed point ($\Delta\beta < 0$, green colour) and regions that do not ($\Delta\beta > 0$, red colour). We observe that effective universality and a UV fixed point is only allowed in two regions: this first is for negative $\mu$ and small $\lambda_3 \approx 0.1$. In fact our fixed-point values for $\mu$ and $\lambda_3$ lie in this region of effective universality. In Fig. 4 it is marked by a white dot. The other region is at positive $\mu$ and large $\lambda_3 > 0.5$. At positive $\mu$ and negative $\lambda_3$ there is another region that allows for effective universality but not for a UV fixed point at positive Newton coupling. Fig. 4 highlights that the common realisation of effective universality and a UV fixed point is highly non-trivial.

As effective universality is restored exactly at the fixed point, i.e. if $\mu$ and $\lambda$ are set to their fixed-point values in $\beta_{G_{(3,0)}}$ and $\beta_{G_{(1,2)}}$, the critical exponents do not reflect effective universality: the real parts of the relevant critical exponents actually feature distinct dependencies on $N_s$, cf. Fig. 5. In particular, the eigendirection which has most overlap with $G_{(1,2)}$ increases in relevance for increasing $N_s$. On the other hand, two superpositions of $G_{(3,0)}$ and $\mu$ form two relevant eigendirections, both of which become less relevant as $N_s$ is increased.

## 4.5 Effective universality in commonly used approximations

It is useful to investigate in which commonly used approximation effective universality survives. We have already observed that in our truncation a derivative expansion spoils effective universality for the $N_s$-independent part, cf. Fig. 2, and it also does not hold anymore away from the fixed point, cf. Fig. 4. We now take a closer look at the $N_s$-dependent part, where from the left panel of Fig. 3 we already infer that a derivative expansion works rather well.

At $\mu = 0 = \lambda_3$ and evaluated with a derivative expansion at $p^2 = 0$, the flow equations for the dynamical Newton couplings are given by the analytic expressions

$$\beta_{G_{(3,0)}} = (2+3\eta_h)G_{(3,0)} - \frac{833}{285\pi}G_{(3,0)}^2 - \frac{43}{570\pi}N_s\, G_{(3,0)}^{1/2}G_{(1,2)}^{3/2}\,,$$

$$\beta_{G_{(1,2)}} = (2+\eta_h+2\eta_\varphi)G_{(1,2)} - \frac{4}{\pi}G_{(1,2)}^2 + \frac{8}{3\pi}G_{(3,0)}^{1/2}G_{(1,2)}^{3/2}\,, \tag{31}$$

where we again identified $G_{(n\geq 3,0)} = G_{(3,0)}$ and $G_{(n,m\geq 2)} = G_{(1,2)}$ and set the anomalous dimension on the right-hand side of the Wetterich equation to zero. The graviton anomalous dimension in the canonical term gives a leading-order contribution and thus we need to include its $N_s$-dependence. With a projection at $p^2 = 0$ the contribution is

$$\eta_h\Big|_{N_s} = \frac{1}{24\pi}N_s\, G_{(1,2)}\,. \tag{32}$$

A projection at finite momentum changes the size of the contribution but not the sign. The scalar anomalous dimension does not have any $N_s$-dependence. We use this together with (31) and find

$$\beta_{G_{(3,0)}}\Big|_{N_s} \approx 0.016\, G^2 N_s\,,$$

$$\beta_{G_{(1,2)}}\Big|_{N_s} \approx 0.013\, G^2 N_s\,. \tag{33}$$

Note that all the above $N_s$-dependent contributions are gauge independent as they originate from scalar loops. We find a good agreement between the two fluctuation couplings $\beta_{G_{(3,0)}}|_{N_s}$ and $\beta_{G_{(1,2)}}|_{N_s}$, having only a 16% deviation of their coefficients.

This computation emphasises that the scalar contribution to different avatars of fluctuation Newton couplings features a semi-qualitative agreement already in simple truncations. This is remarkable, as the terms from the anomalous dimension $\eta_h$ differ by a factor three and the agreement originates in a subtraction of the anomalous dimension and the vertex flow in $\beta_{G_{(3,0)}}$. Moreover, the behaviour is in agreement with that of $G_{(1,2)}$ in [8], where a different choice of parameterisation and gauge was employed. This robustness with respect to (unphysical) variations of the scheme is encouraging.

## 5 Effective universality for the background-fluctuation system

We now address our second key question and explore whether effective universality is also present at the level of the background Newton coupling. We first investigate this in the full system where the fluctuation couplings drive the flow of the background couplings and then we turn to the background field approximation and hybrid fluctuation-background–computations.

### 5.1 Effective universality for the background Newton coupling

The flow of $\bar{G}$ and $\bar{\lambda}$ is driven exclusively by the fluctuation couplings, their flow equations can be found in App. A. Thus we simply insert the fixed-point value for $\mu$ and the fluctuation-

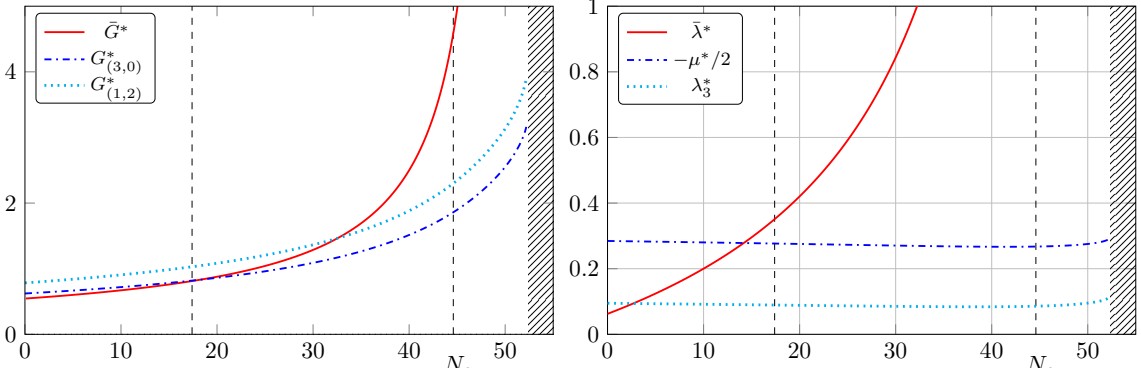

Figure 6: Displayed are the fixed-point values as a function of $N_s$. We use the fluctuation system as input on the right-hand side of the Wetterich equation. Left: Fixed-point values of background and fluctuation Newton couplings. Right: Fixed-point values of background and fluctuation cosmological constant. The vertical lines at $N_s \approx 17.5$ and $N_s \approx 44.6$ show where $\eta_h(0)$ and $\eta_h(k^2)$ exceed two, respectively.

field anomalous dimensions on the right-hand side of the Wetterich equation, evaluated at $G^*_{(3,0)}, G^*_{(1,2)}, \mu^*$ and $\lambda^*_3$.

As a function of $N_s$, we observe fixed-point values for $\bar{G}^*$ that track those of the fluctuation system at the qualitative level, cf. the left panel of Fig. 6. A similar conclusion can be drawn from the one-loop $\beta$-function for $\bar{G}$, which reads

$$\beta_{\bar{G}} = 2\bar{G} - (3.64 - 0.057 N_s) \bar{G}^2. \tag{34}$$

Here we have evaluated the $\bar{G}^2$ coefficient on $\mu^*$, $\eta^*_h(p^2 = k^2)$ and $\eta^*_c(p^2 = k^2)$ at $N_s = 1$, thereby neglecting an additional $N_s$-dependence from the fixed-point couplings themselves. The pure-gravity coefficient differs by 6% in comparison to $\beta_{G_{(3,0)}}$, cf. (28), and the $N_s$-dependent coefficient by a factor 4.4. Both signs agree with those in the fluctuation system. The substantial deviation of the $N_s$-dependent coefficient leads to a larger gap between the fluctuation and the background avatars at large $N_s$. Nevertheless, the $N_s$-dependent fixed-point values for all avatars of the Newton coupling agree on the qualitative level.

The fixed-point value of the background coupling $\bar{\lambda}$ differs significantly from its fluctuation version, cf. the right panel of Fig. 6. Importantly, $\bar{\lambda}$ and $\lambda_3$ can cross the value $\frac{1}{2}$, while the graviton mass parameter $\mu$ cannot cross the pole located at $\mu = -2\lambda_2 = -1$. As already discussed in the beginning of Sec. 4, we do not expect effective universality for those couplings due to the special rôle of $\mu$. This again emphasises that at least $\mu$ should be taken from a fluctuation computation in order to obtain robust results.

## 5.2 The fate of effective universality in commonly used approximations

### 5.2.1 Background field approximation

While it is the fluctuation-field propagator that drives the flow, we are ultimately interested in the background effective action $\Gamma_{k \to 0}[\bar{g}_{\mu\nu} = g_{\mu\nu}, h_{\mu\nu} = 0]$ to read off the physics. A commonly used approximation is thus the background field approximation, which consists in inserting $\Gamma_k^{(2,0)}[\bar{g}_{\mu\nu} = g_{\mu\nu}, h_{\mu\nu} = 0]$ on the right-hand side of the Wetterich equation and thereby letting the background couplings drive the flow, see also (9) and the next section for details. Although this leads to a semi-quantitative agreement with the full results at $N_s = 0$, the approximation fails to capture even the qualitative $N_s$ dependence correctly. While this might in principle

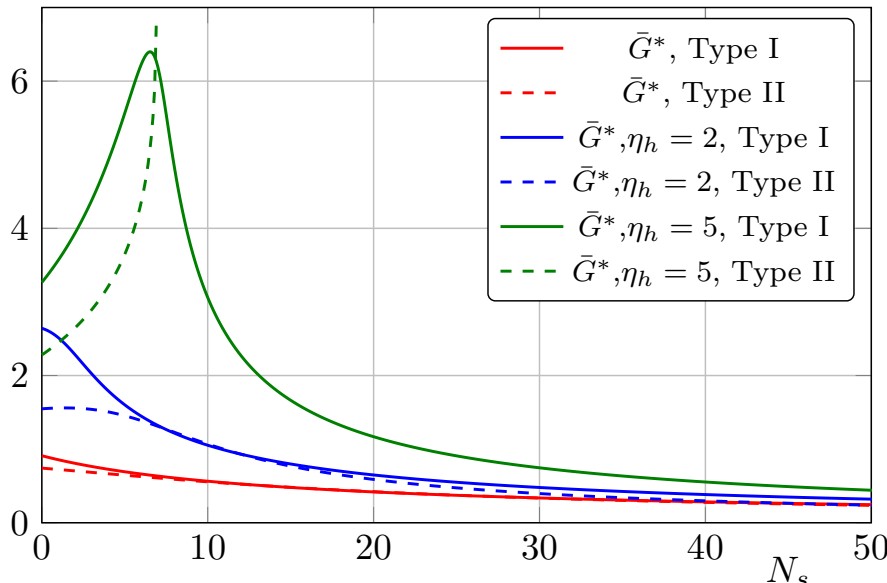

Figure 7: Fixed-point values of the background Newton coupling for a type-I (continuous lines) and a type-II regulator (dashed lines) with the background-field approximation ($\eta_h = -2$, red lines), and in hybrid cases with $\eta_h = 2$ (blue lines) and $\eta_h = 5$ (green lines). See App. A for the definitions of type-I and -II regulators.

improve in extended approximations, it casts some doubt on the use of the background-field approximation for gravity-matter systems at least in the case of scalar matter.

For the background Newton coupling, the flow equation in the background-field approximation, i.e., at $\eta_h = -2$, evaluated at $\bar{\lambda} = 0$ reads

$$\beta_{\bar{G}} = 2\bar{G} - \left(\frac{79}{4} - N_s\right)\frac{\bar{G}^2}{6\pi} \approx 2\bar{G} - (1.05 - 0.053 N_s)\bar{G}^2, \tag{35}$$

where the signs of the coefficients still agree with those of the fluctuation system. The failure of the background-field approximation to correctly capture the $N_s$ dependence in our truncation is a consequence of the difference between $\mu$ and $\bar{\lambda}$. While $\mu^*$ stays approximately constant with increasing $N_s$, $\bar{\lambda}^*$ is driven towards larger values, thereby enhancing gravity fluctuations and suppressing the effect of scalar-matter fluctuations. The background cosmological constant with fluctuation input is displayed in the right panel of Fig.6. The growth of $\bar{\lambda}^*$ with increasing $N_s$ remains the same in the background-field approximation, but it approaches $\bar{\lambda}^* = \frac{1}{2}$ asymptotically, as it cannot cross this pole in this approximation.

### 5.2.2 Hybrid scheme for $\eta_h$

In a hybrid scheme, put forward in [3, 44] and employed in the analysis of gravity-matter systems in [70], the graviton anomalous dimension is distinguished from the anomalous dimension of the background Newton coupling. While $\eta_h$ is evaluated as a function of the background couplings $\bar{G}$ and $\bar{\lambda}$ in this hybrid, it can deviate from the background-value $\eta_h = -2$, and thereby partially account for the nontrivial anomalous dimension of the graviton. In particular, $\eta_h$ matches that of a fluctuation computation as an expression of $G$ and $\lambda$; the difference arises from the insertion of different fixed-point values.

Within such a hybrid setup, a behaviour qualitatively closer to that of the full fluctuation system was observed, [70], i.e. the fixed-point value for Newton coupling rose as a function of $N_s$. This can be traced back to a growth of the anomalous dimension, cf. Fig.7. A strong growth

of the anomalous dimension has to be considered carefully: the usual choice of regulators is $R_k \sim Z_h$, implying a bound on the anomalous dimensions $\eta < 2$ (for bosonic fields) [7]. As $Z_h \sim k^{-\eta_h}$ in the fixed-point regime, $\eta_h > 2$ destroys the UV behaviour of the regulator that should suppress all modes in the limit $k \to \infty$. For $\eta_h > 4$ signs of diagrams in $\beta$-functions start to flip. Furthermore a large anomalous dimension can be interpreted as a hint at large relative cutoff scales between the different fields of the theory [10].

We demonstrate the transition between the $N_s$ dependence of $\bar{G}^*$ in the strict background-field approximation and the hybrid scheme by setting the anomalous dimension to fixed successively increasing values, cf. Fig. 7. We investigate type-I and type-II regulators [35], see also App. A for their definitions. Indeed the background Newton coupling with a type-II regulator rises as soon as $\eta_h > 4$ and the diagrams in the $\beta$-function have flipped its sign. For the type-I regulator this happens at $\eta_h > 6$, as it features higher powers of $(1 - 2\lambda)$ in the denominator, which flip their sign only at $\eta_h > 6$. The fixed-point results for the Newton coupling in the hybrid scheme therefore match qualitatively with the effectively universal results of the fluctuation system.

## 6 Level-one improvement

In Sec. 4 we found effective universality in the (quenched) fluctuation system. This effective universality is not present in the background system, which only shows a qualitative, not a quantitative agreement. This motivates us to investigate at which order of the fluctuation computation effective universality sets in. In particular, in (8) we have presented a split of the scale dependent effective action in a diffeomorphism-invariant part and a gauge part. Effective universality hints at a form of the effective action such that

$$\Gamma_k[\bar{g}_{\mu\nu}, h_{\mu\nu}, \varphi] = \Gamma_k^{\text{diff}}[g_{\mu\nu}, \varphi] + \Delta\Gamma_k^{\text{gauge}}[\bar{g}_{\mu\nu}] + h_{\rho\sigma}\Delta\Gamma_{k,\rho\sigma}^{\text{gauge}}[\bar{g}_{\mu\nu}], \tag{36}$$

i.e. an effective action where the gauge part is only linear in the fluctuation field. The higher orders are sub-leading.

In this section we upgrade the background couplings to level-one couplings with the use of Nielsen identities. By comparing the level-one couplings to fluctuation ones, we specifically test the importance of the term linear in $h_{\mu\nu}$ in (36), i.e. $\Delta\Gamma_{k,\rho\sigma}^{\text{gauge}}$.

Furthermore, it is desirable to find simple approximations of the system that ideally do not require the separate calculation of both background and fluctuation flows. Therefore we study whether the background-field approximation, upgraded to a level-one system by using the modified split Ward identity, can qualitatively or even quantitatively reproduce the behaviour of the full system.

### 6.1 Nielsen or split Ward identity and its applications

We exploit the Nielsen identity (NI) or split Ward identity (sWI) to improve upon the background-field approximation. Related derivations and applications in the present context can be found in [111–115]. Split Ward identities in the context of quantum gravity have also been discussed in [17–27, 116, 117], see also [118].

With the introduction of a background field, the effective action becomes a functional of both the dynamical fluctuation field $\Phi$ and the auxiliary background field $\bar{\Phi}$

$$\Gamma_k = \Gamma_k[\bar{\Phi}, \Phi]. \tag{37}$$

In the present case of gravity coupled to scalar matter the background and fluctuation fields read

$$\bar{\Phi} = (\bar{g}_{\mu\nu}, 0, 0, \bar{\phi}), \qquad \Phi = (h_{\mu\nu}, c_\mu, \bar{c}_\mu, \varphi), \tag{38}$$

respectively, where the full metric and scalar fields are given by

$$g_{\mu\nu} = \bar{g}_{\mu\nu} + h_{\mu\nu}, \quad \phi = \bar{\phi} + \varphi. \tag{39}$$

In scalar theories there is no need to choose the cutoff to depend on the background field. Thus the flowing action is only a function of the full field $\phi = \bar{\phi} + \varphi$. For the purpose of illustration, we introduce a dependence of the cutoff on $\bar{\phi}$ artificially. The effective action at $k = 0$ is a functional of the full field $\phi = \bar{\phi} + \varphi$ only. This is due to the fact that the classical action has this property, $S_{\text{cl}}[\bar{\phi}, \varphi] = S_{\text{cl}}[\bar{\phi} + \varphi]$. The shift symmetry is broken by the cutoff term $R_k = R_k[\bar{\phi}]$. The resulting difference in the dependence on the two fields is sourced only by the cutoff term and expressed by the NI/sWI, [111, 117]

$$\frac{\delta\Gamma_k}{\delta\bar{\phi}} - \frac{\delta\Gamma_k}{\delta\varphi} = \frac{1}{2}\text{Tr}\left[\frac{\delta R_k[\bar{\phi}]}{\delta\bar{\phi}} G_k[\bar{\phi}, \varphi]\right]. \tag{40}$$

This equation is derived in straight analogy to the flow equation itself (6a), which is reflected in the structural similarities [111]. We again use the shorthand $G_k$ for the regularised propagator of the fluctuation field, see (6b). For flows towards the IR where the regulator vanishes, (40) suggests to use the background-field approximation

$$\Gamma_k[\bar{\phi}, \varphi] \approx \Gamma_k[\bar{\phi} + \varphi, \varphi = 0], \tag{41}$$

which is exact for $k = 0$ in the present example of scalar theories. However, for flows towards the UV ($k \to \infty$) the background-field approximation is spoiled by power-counting leading terms in the effective action. This is a consequence of the mass-like nature of the cutoff, which makes it power-counting UV relevant. In the following, we do not include the scalar background field into the regulator, thus the right-hand side of (40) is zero and we focus on the NI for the graviton.

In gravity, and gauge theories in general, the situation is more complicated. In this case there are two sources for the background dependence of the effective action. In addition to the regulator term, the second source of background dependence comes from the gauge fixing sector $S_{\text{gauge}} = S_{\text{gf}} + S_{\text{gh}}$. Both the gauge fixing term $S_{\text{gf}}$ and the ghost term $S_{\text{gh}}$ have to depend on the background field if background gauge invariance is demanded. Thus the motivation for introducing the background field, namely background gauge invariance, leads to a genuine background dependence of the effective action. For our gravity-matter system the NI (40) turns into

$$\frac{\delta\Gamma_k}{\delta\bar{g}_{\mu\nu}} - \frac{\delta\Gamma_k}{\delta h_{\mu\nu}} = \frac{1}{2}\text{Tr}\left[\frac{1}{\sqrt{\bar{g}}}\frac{\delta\sqrt{\bar{g}}R_k[\bar{g}]}{\delta\bar{g}_{\mu\nu}} G_k[\bar{g}, h]\right] + \left\langle\frac{\delta S_{\text{gauge}}[\bar{g}, h]}{\delta\bar{g}_{\mu\nu}} - \frac{\delta S_{\text{gauge}}[\bar{g}, h]}{\delta h_{\mu\nu}}\right\rangle, \tag{42}$$

where the regulator $R_k$ is now a matrix in field space. The second line in (42) originates from the background-field dependence of the gauge fixing sector, and it survives in the limit $k \to 0$ if the effective action is evaluated off-shell, but vanishes on the solution of the equations of motion [12, 119]. In the present work we approximate (42) and use

$$\lim_{k\to\infty}\left(\frac{\delta\Gamma_k}{\delta\bar{g}_{\mu\nu}} - \frac{\delta\Gamma_k}{\delta h_{\mu\nu}}\right) \simeq \frac{1}{2}\text{Tr}\left[\frac{1}{\sqrt{\bar{g}}}\frac{\delta\sqrt{\bar{g}}R_k[\bar{g}]}{\delta\bar{g}_{\mu\nu}} G_k[\bar{g}, h]\right], \tag{43}$$

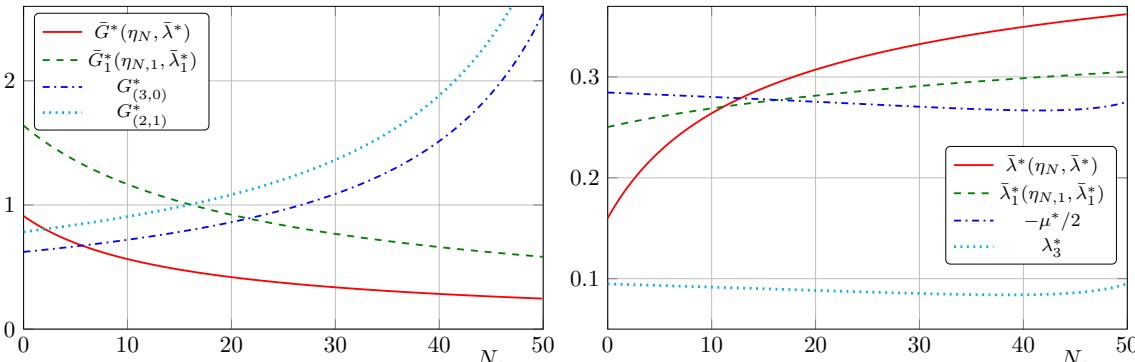

Figure 8: Fixed point values of the avatars of the Newton coupling (left panel) and the cosmological constant (right panel) as a function of $N_s$ in the background-field approximation (red continuous lines), in the level-one approximation (green dashed lines) and for the fluctuation system (blue dot-dashed and light blue dotted lines).

where we have dropped the second line with the gauge fixing contributions. Two arguments underlie our choice to focus on the cutoff term: First, at the present level of truncation it is actually possible to effectively subsume changes in the gauge fixing under changes of the regulator: Specifically we concentrate on momenta $p^2 \lesssim k^2$ and a given gauge fixing $S_{\text{gf}}^{(2)}(p^2)$. In this regime we can utilise the generality of the regulator to effectively re-adjust it

$$R_k \rightarrow R_k - S_{\text{gf}}^{(2)}(p^2)\, r(p^2/k^2) + S_{\text{gf,diff}}^{(2)}(p^2)\, r(p^2/k^2)\,, \tag{44}$$

where $S_{\text{gf,diff}}^{(2)}$ is a general gauge fixing term. Hence, with (44) we have effectively changed the gauge fixing term for momenta $p^2 \lesssim k^2$. If applying this procedure to the ghost, it is only possible to change its propagator and the interaction of the ghost with the background graviton $\bar{g}_{\mu\nu}$, but not that with the dynamical graviton $h_{\mu\nu}$. As the ghost terms do not take a leading rôle in the flows, this is negligible. In the background-field approximation, and using the standard expansion in powers of the curvature, the above mapping strictly holds. In summary, for the study of different gauge fixing terms in the present approximation it suffices to study the regulator dependence of the flow for momenta $p^2 \lesssim k^2$. In this work we refrain from exploiting this freedom in practice.

Second, if one compares the contributions of the cutoff term and the gauge fixing sector to the UV flow, a counting argument suggests that the cutoff term dominates. This is because it couples to all fluctuation modes of the graviton, while the contributions of the gauge fixing sector couple directly only to the longitudinal modes. Hence, the transverse-traceless approximation, which focuses on the spin-2 mode of the graviton, is only affected by the regulator term (43).

Finally we are interested in the relation between background and fluctuation field two-point functions. To that end we apply $(\delta/\delta\bar{g}_{\rho\sigma} + \delta/\delta h_{\rho\sigma})$ to (43) and drop the cross terms as further approximation, which yields

$$\frac{\delta^2 \Gamma_k}{\delta h_{\rho\sigma} \delta h_{\mu\nu}} - \frac{\delta^2 \Gamma_k}{\delta \bar{g}_{\rho\sigma} \delta \bar{g}_{\mu\nu}} \simeq -\frac{1}{2}\left(\frac{\delta}{\delta \bar{g}_{\rho\sigma}} + \frac{\delta}{\delta h_{\rho\sigma}}\right)\text{Tr}\left[\frac{1}{\sqrt{\bar{g}}}\frac{\delta \sqrt{\bar{g}} R_k[\bar{g}]}{\delta \bar{g}_{\mu\nu}} G_k[\bar{g}, h]\right]. \tag{45}$$

In the example of Yang-Mills theory [111, 112], the analogous fluctuation field derivative of the term in the square brackets on the right-hand side of (45) gives sub-leading contributions. We test a similar assumption and thereby arrive at the final approximation for the fluctuation

two-point function, which we use in order to close the flow equation,

$$\frac{\delta^2 \Gamma_k}{\delta h_{\mu\nu} \delta h_{\rho\sigma}} \approx \frac{\delta^2 \Gamma_k}{\delta \bar{g}_{\mu\nu} \delta \bar{g}_{\rho\sigma}} - \frac{1}{2} \frac{\delta}{\delta \bar{g}_{\rho\sigma}} \text{Tr} \left[ \frac{1}{\sqrt{\bar{g}}} \frac{\delta \sqrt{\bar{g}} R_k}{\delta \bar{g}_{\mu\nu}} G_k \right]. \tag{46}$$

This approximation has been used in gravity in [20, 116]. Apart from the standard background two-point function it contains a second, regulator-induced term. Importantly, it can be straightforwardly computed with heat-kernel methods, see App. C for details.

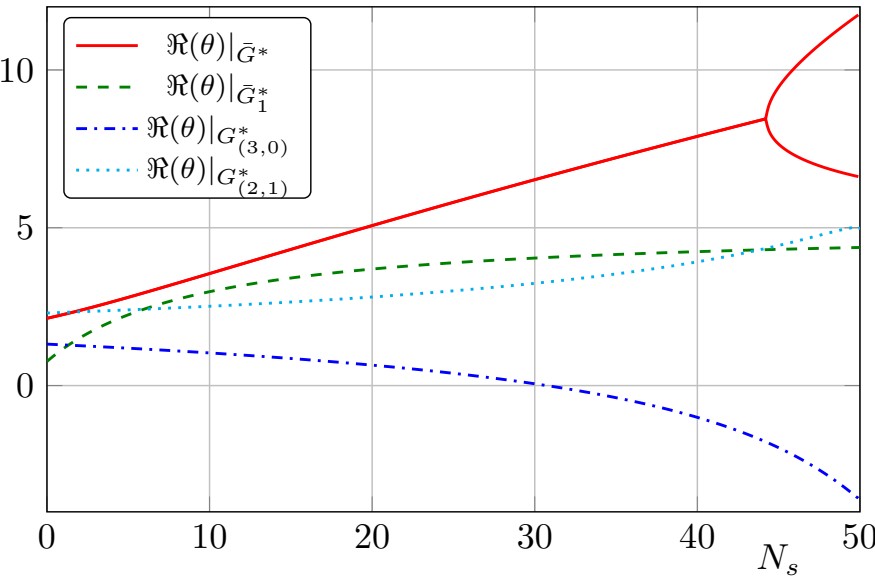

Figure 9: Real part of the relevant critical exponents as a function of $N_s$ in the background-field approximation (red continuous line), in the level-one approximation (green dashed line) and for the fluctuation system (blue dot-dashed and light blue dotted lines).

## 6.2 Fixed-point results for level-one couplings

Our procedure provides us with a set of $\beta$-functions for the dimensionless level-one couplings, $\bar{G}_1$ and $\bar{\lambda}_1$, which are displayed in App. B. We now analyse whether the level-one improvement leads to a system that reproduces the fluctuation results more closely than the background-field approximation.

In Fig. 8 we display the fixed-point values of the fluctuation, the level-one, and the background system. The input on the right-hand side of the Wetterich equation is fluctuation, level-one, and background couplings, respectively. In Fig. 9 we present the corresponding real parts of the relevant critical exponents. The background and the level-one system each contain exactly two couplings. Both of them are relevant and their associated critical exponents form a complex conjugated pair. The fluctuation system has four dynamical couplings and four non-dynamical background couplings (background and level-one couplings). Of the four dynamical couplings three are relevant and one is irrelevant. Two relevant critical exponents form a complex conjugated pair and their real part is displayed in Fig. 9 since one can associate them with the couplings $\mu$ and $G_{(3,0)}$ by means of the largest overlap of the corresponding eigenvector. In Fig. 10 we display the fixed-point values of the background, level-one, and fluctuation couplings with the full fluctuation system as input on the right-hand side of the Wetterich equation.

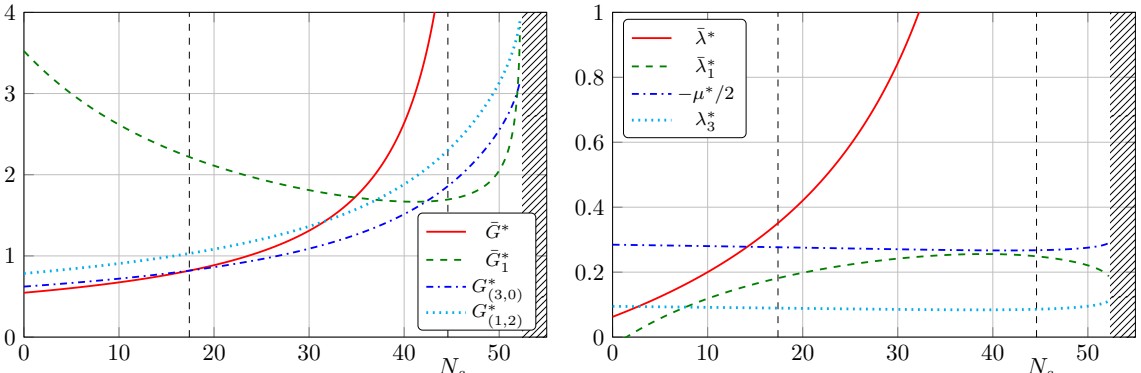

Figure 10: Comparison of background, level-one and fluctuation avatars of the Newton coupling (left panel) and the cosmological constant (right panel). All couplings are evaluated with the input of the full fluctuation system on the right-hand side. The vertical lines at $N_s \approx 17.5$ and $N_s \approx 44.6$ show where $\eta_h(0)$ and $\eta_h(k^2)$ exceed two, respectively.

We observe that in pure gravity, the level-one improvement leads to critical exponents that agree better with the fluctuation results than the background-field approximation, cf. Fig. 9. As a function of $N_s$, the fluctuation results show a qualitatively different behaviour than the background and the level-one results. The real parts of critical exponents of the latter are increasing as a function of $N_s$ while they are decreasing in the fluctuation case and even become irrelevant at $N_s \approx 31$. Nevertheless the level-one critical exponents are growing slower than the background ones and thus we observe a slight improvement.

For the fixed-point values of the Newton couplings the level-one approximation tracks the background-approximation in its qualitative dependence on $N_s$: they are both decreasing as a function of $N_s$, while the fluctuation Newton couplings are increasing. However the quantitative difference between $G^*_{(3,0)}$ and $\bar{G}^*_1$ is smaller than the quantitative difference between $G^*_{(3,0)}$ and $\bar{G}^*$, cf. Fig. 8. In the sector of the cosmological constants or momentum-independent parts of the $n$-point functions, the level-one improvement is more pronounced. While the background cosmological constant increases strongly and approaches the pole at $\bar{\lambda} = \frac{1}{2}$, the level-one coupling remains almost constant and increases only slightly. The fluctuation coupling $-\mu/2$ also remains almost constant but decreases slightly with $N_s$.

In summary the level-one approximation might be considered a slight improvement over the background-field approximation. We have observed slight improvements in the critical exponents and in the fixed-point values of the Newton couplings and the cosmological constants. Considering however its failure to adequately capture the fluctuation results, a level-one approximation seems hardly justified in view of the significantly increased computational effort—at least based on the results in our truncation.

Last but not least we consider the fixed-point results when all couplings, including the background and the level-one coupling, are evaluated with the fluctuation couplings on the right-hand side of the Wetterich equation, cf. Fig. 10. We observe that the level-one approximation even appears to break the effective universality that was observed for the Newton coupling: the qualitative and quantitative dependence of $\bar{G}^*_1$ on $N_s$ does not match that of the other couplings as it first decreases with $N_s$ and then increases strongly. For the 'cosmological constants', we make the opposite observation: while the background cosmological constant deviates strongly from the $N_s$-dependence of the graviton mass parameter, the level-one cosmological constant approaches it towards larger $N_s$. For the canonically most relevant coupling in the truncation, the step from the background coupling to the level-one coupling is therefore a significant step towards capturing the behaviour of the fluctuation coupling.

# 7 Summary and outlook

In this work we have focused on two key questions within the asymptotic safety program for quantum gravity.

Firstly, we have introduced and explored the concept of *effective universality* for the dynamical couplings in matter-gravity systems. Gauge theories always feature several avatars of the same coupling, such as, e.g. the quark-gluon coupling and the three-gluon and four-gluon coupling in QCD. These couplings are related by gauge invariance. Together with the marginal nature of these couplings in $d = 4$ and the corresponding two-loop universality this offers a universal definition of the gauge coupling.

In gravity in $d = 4$, the Newton coupling is dimensionful and a similar form of universality is therefore not to be expected. Different avatars of the Newton coupling, defined, e.g. from the three-graviton vertex and the scalar-graviton vertex, are therefore by no means guaranteed to agree. On the other hand, diffeomorphism invariance of course implies a relation between those couplings. We define *effective universality* as a quantitative agreement of these couplings, at least at the asymptotically-safe fixed point. In this work, the viability of this concept has been explored in a scalar-gravity-system with $N_s$ scalars. We have specifically focused on the scalar-graviton coupling and the three-graviton coupling. We have discovered indications for effective universality in the quenched approximation, where closed scalar loops are neglected. Apart from a difference between the two Newton couplings at small momenta, they show a remarkable quantitative agreement, in particular in the most relevant range of momenta close to the cutoff. "Unquenching" quantum gravity leads to $N_s$-dependent corrections which are subleading in the tentative range of validity of our truncation. While effective universality in the sense of a quantitative agreement is lost, we still observe qualitative agreement of the contributions. We have also discussed effective universality in commonly used approximations, ranging from bilocal schemes to the derivative expansion. We find that effective universality even holds for the derivative expansion of the flows about vanishing momentum, despite the large quantitative deviations from the full flows. Obtaining quantitative results requires going beyond the derivative expansion using at least bilocal schemes.

Our results provide a justification for a commonly used approximation, in which different avatars of the Newton coupling are equated. In terms of exploring the consequences of asymptotically safe quantum gravity within efficient, i.e. small, truncations, effective universality is key.

Moreover, the emergence of effective universality at the fixed point is a very strong indication for the physical nature of the asymptotically-safe fixed point, in the following sense: For a truncation-induced fixed point, there is no reason to exhibit effective universality, since it is simply a "random" solution to a set of polynomial equations. On the other hand, for an actual fixed point, diffeomorphism invariance provides a mechanism to restore effective universality at the fixed point. In particular, in our study, even the qualitative agreement of the beta functions for different avatars of the Newton coupling in the unquenched setting requires non-trivial cancellations between different contributions to the beta functions. In our opinion these cancellations are extremely unlikely to randomly occur in truncation-induced fixed points.

We then have proceeded to tackle the second key question, namely the relation of the dynamical system to the background system. We have focused on a search for effective universality, and discovered a qualitative agreement of the beta function for the background Newton coupling with that of the dynamical system. However, effective universality is lost, casting doubt on the use of the background approximation for quantitatively reliable results. Moreover, the beta functions for the cosmological constant and graviton mass parameter are manifestly different, leading to different behaviour in the two systems at large $N_s$ in our truncation.

In practise, our results can be read as tentatively suggesting that at least the graviton mass parameter should always be taken from a fluctuation calculation.

Finally, we explore whether an upgrade of the background system to a level-one system by means of the modified Nielsen or shift Ward identity restores effective universality. Focusing on the regulator contribution to the Nielsen identity, we find that the level-one upgrade is insufficient. Still, interestingly the level-one cosmological constant is the only coupling that significantly improves towards the corresponding fluctuation result for the graviton mass parameter.

In summary, our results yield an a-posteriori justification for the use of relatively simple truncations that make use of effective universality. Even more important, the highly nontrivial signatures for effective universality in scalar-gravity systems provide further evidence for asymptotic safety in quantum gravity.

**Acknowledgements** We thank N. Christiansen, S. Lippoldt and R. Percacci for discussions. AE is supported by an Emmy-Noether-grant of the DFG under AE/1037-1. PL thanks the ITP Heidelberg for hospitality during a research stay funded by the DFG grant AE/1037-1. MR acknowledges funding from IMPRS-PTFS. This work is supported by EMMI and is part of and supported by the DFG Collaborative Research Centre "SFB 1225 (ISOQUANT)".

# A    Background flow equations

In this Appendix we display the background-flow equations. We use a Litim-type regulator [109, 110] for all fields in our setup

$$R_k^{ij}(p^2) = \delta^{ij} \ \Gamma^{(\phi_i \phi_i^*)}(p^2)\big|_{\mu=0} \, r_{\phi_i}(p^2/k^2),$$
$$r(x) = \left(\frac{1}{x} - 1\right)\theta(1-x). \tag{47}$$

We employ the gauge $\alpha = \beta = 0$ and use the Laplacian as coarse-graining operator, i.e., a type-I regulator. In Fig. 7 we also used a type-II regulator for comparisons, which means that we used the Laplacian plus explicit curvature terms of the two-point functions as coarse graining operator, see [35] for more details on the different types of regulators. We further employ the York-decomposition [103, 104] for the graviton

$$h_{\mu\nu} = h_{\mu\nu}^{\text{tt}} + \frac{1}{d}\bar{g}_{\mu\nu}h^{\text{tr}} + 2\bar{\nabla}_{(\mu}\xi_{\nu)} + \left(\bar{\nabla}_\mu\bar{\nabla}_\nu - \frac{\bar{g}_{\mu\nu}}{d}\bar{\nabla}^2\right)\sigma, \tag{48}$$

and the ghost

$$c_\mu = c_\mu^{\text{T}} + \bar{\nabla}_\mu\eta, \tag{49}$$

with field redefinitions according to [28, 51, 64]

$$\xi^\mu \to \frac{1}{\sqrt{\bar{\Delta} - \frac{\bar{R}}{4}}}\xi^\mu,$$
$$\sigma \to \frac{1}{\sqrt{\bar{\Delta}^2 - \bar{\Delta}\frac{\bar{R}}{3}}}\sigma,$$
$$\eta \to \frac{1}{\bar{\Delta}}\eta. \tag{50}$$

The result for the gravity part of the background flows agrees with e.g. [6, 51] and the scalar part agrees with e.g. [70]. The details of the computation are given in App. C. The flow of the background couplings are given by

$$\partial_t \bar{G} = (2 + \eta_N)\bar{G}\,,$$

$$\partial_t \bar{\lambda} = -4\bar{\lambda} + \frac{\bar{\lambda}}{\bar{G}}\partial_t \bar{G} + 8\pi\bar{G}\text{Flow}_{\bar{\Gamma}}\bigg|_{\sqrt{\bar{g}}\text{-terms}}\,,$$

$$\eta_N = 16\pi\bar{G}\text{Flow}_{\bar{\Gamma}}\bigg|_{\sqrt{\bar{g}}\bar{R}\text{-terms}}\,, \tag{51}$$

where

$$\text{Flow}_{\bar{\Gamma}} = \frac{\sqrt{\bar{g}}}{32\pi^2}\left\{\left[\frac{5 - \frac{5}{6}\eta_h}{1 - 2\lambda} + \frac{1 - \frac{1}{6}\eta_h}{1 - \frac{4}{3}\lambda} - 4 - \frac{2}{3}\eta_h + \frac{4}{3}\eta_c + N_s(1 - \frac{1}{6}\eta_\varphi)\right]k^4\right.$$

$$\left. + \left[-\frac{10}{3}\frac{1 - \frac{1}{6}\eta_h}{(1 - 2\lambda)^2} - \frac{5}{3}\frac{1 - \frac{1}{4}\eta_h}{1 - 2\lambda} + \frac{1}{3}\frac{1 - \frac{1}{4}\eta_h}{1 - \frac{4}{3}\lambda} - \frac{23}{12} - \frac{7}{18}\eta_h + \frac{7}{9}\eta_c + \frac{1}{3}N_s(1 - \frac{1}{4}\eta_\varphi)\right]k^2\bar{R}\right\}$$

$$+ \mathcal{O}(\bar{R}^2)\,. \tag{52}$$

Note that the quantities $\lambda$, $\eta_h$, $\eta_c$, and $\eta_\varphi$ can be taken from the respective fluctuation two-point functions. In this case the background couplings are non-dynamical spectators. The usual background-field approximation is obtained by setting $\lambda = \bar{\lambda}$, $\eta_h = \eta_N$, $\eta_c = 0$, and $\eta_\varphi = 0$.

## B    Level-one flow equations

In this Appendix we display the level-one flow equations that are derived though a Nielsen identity from the background-flow equations, see Sec. 6. We work with the approximation

$$\frac{\delta}{\delta h_{\mu\nu}}\partial_t\Gamma_k \approx \frac{\delta}{\delta \bar{g}_{\mu\nu}}\partial_t\Gamma_k - \partial_t\left(\frac{1}{2}\text{Tr}\left[\frac{1}{\sqrt{\bar{g}}}\frac{\delta\sqrt{\bar{g}}R_k}{\delta\bar{g}_{\mu\nu}}G_k\right]\right)\,. \tag{53}$$

Consequently we are interested in evaluating

$$\mathscr{I}^{\mu\nu} = I\,\bar{g}^{\mu\nu} = \frac{1}{2}\text{Tr}\left[\frac{1}{\sqrt{\bar{g}}}\frac{\delta\sqrt{\bar{g}}R_k}{\delta\bar{g}_{\mu\nu}}G_k\right]\,, \tag{54}$$

which gives us, combined with the background flows, the flows for the level-one couplings. The trace appearing in (54) can be evaluated using heat kernel techniques. Details of the computation are presented in App. C. The result for $I$ is given by

$$8I = \frac{\sqrt{\bar{g}}}{32\pi^2}\left\{\left[\frac{10}{3}\frac{1}{1 - 2\lambda} + \frac{2}{3}\frac{1}{1 - \frac{4}{3}\lambda} - \frac{8}{3} + \frac{2}{3}N_s\right]k^4\right.$$

$$\left. + \left[-\frac{20}{9}\frac{1}{(1 - 2\lambda)^2} - \frac{5}{2}\frac{1}{1 - 2\lambda} + \frac{1}{2}\frac{1}{1 - \frac{4}{3}\lambda} - \frac{71}{36} + \frac{1}{2}N_s\right]k^2\bar{R}\right\} + \mathcal{O}(\bar{R}^2)\,. \tag{55}$$

We display the result for $8I$, since the Nielsen identity enters precisely with this factor in the flow equation for the $\sqrt{\bar{g}}$-terms as well as the $\sqrt{\bar{g}}\bar{R}$-terms. The reason for this is that $\mathscr{I}^{\mu\nu}$

enters with a $\partial_t$ derivative, $\partial_t \sqrt{\bar{g}} k^4 = 4\sqrt{\bar{g}} k^4$ and $\partial_t \sqrt{\bar{g}}\bar{R}k^2 = 2\sqrt{\bar{g}}\bar{R}k^2$, while $\text{Flow}_{\bar{\Gamma}}$ enters with a $\delta_{\bar{g}}$ derivative, $\delta_{\bar{g}} \sqrt{\bar{g}} = \frac{1}{2}\sqrt{\bar{g}}\bar{g}^{\mu\nu}$ and $\delta_{\bar{g}} \sqrt{\bar{g}}\bar{R} = \frac{1}{4}\sqrt{\bar{g}}\bar{g}^{\mu\nu}\bar{R}$ cf. (66). Consequently in both cases they combine to the factor 8 as indicated above. Note that $\mathscr{I}^{\mu\nu}$ does not contribute to the flow equation for $\sqrt{\bar{g}}\bar{R}^2$ since $\partial_t \sqrt{\bar{g}}\bar{R}^2 k^0 = 0$. This is expected since $R^2$ and $R^2_{\mu\nu}$ are marginal couplings and thus their one-loop flow equations are universal. Note furthermore that in the above discussion we have neglected terms like $\partial_t \lambda$. Such terms do not change the fixed-point values but they can influence the critical exponents.

The flow equations for the level-one couplings can now be expressed by flow of the background couplings plus the improvement from the Nielsen identity, to wit

$$\partial_t \bar{G}_1 = (2 + \eta_{N,1})\bar{G}_1 \,,$$

$$\partial_t \bar{\lambda}_1 = -4\bar{\lambda}_1 + \frac{\bar{\lambda}_1}{\bar{G}_1}\partial_t \bar{G}_1 + 8\pi \bar{G}_1 \left(\text{Flow}_{\bar{\Gamma}} - 8I\right)\bigg|_{\sqrt{\bar{g}}\text{-terms}} \,,$$

$$\eta_{N,1} = 16\pi \bar{G}_1 \left(\text{Flow}_{\bar{\Gamma}} - 8I\right)\bigg|_{\sqrt{\bar{g}}\bar{R}\text{-terms}} \,. \tag{56}$$

Again, the quantities $\lambda$, $\eta_h$, $\eta_c$, and $\eta_\varphi$ can be taken from the respective fluctuation two-point functions. Then the level-one couplings are non-dynamical spectators. Otherwise we can close the equation at the level-one couplings by setting $\lambda = \bar{\lambda}_1$, $\eta_h = \eta_{N,1}$ and $\eta_c = \eta_\varphi = 0$. The latter is an improved background-field approximation.

## C Evaluation of traces

In this Appendix we present the computation of the Nielsen identity from subsection 6.1. The computation of the trace-term in (46) is the challenging part. For this we first expand the propagator in orders of background curvature

$$\mathscr{I}^{\mu\nu} \equiv I\,\bar{g}^{\mu\nu} := \frac{1}{2}\text{Tr}\left[\frac{1}{\sqrt{\bar{g}}}\frac{\delta\sqrt{\bar{g}}R_k}{\delta\bar{g}_{\mu\nu}}G_k\right]$$

$$= \frac{1}{4}\bar{g}^{\mu\nu}\text{Tr}\left[R_k G_k\right] + \frac{1}{2}\text{Tr}\left[\frac{\delta R_k}{\delta\bar{g}_{\mu\nu}}G_k(\bar{R}=0)\right]$$

$$+ \frac{1}{2}\bar{R}\,\text{Tr}\left[\frac{\delta R_k}{\delta\bar{g}_{\mu\nu}}G_k'(\bar{R}=0)\right] + \mathcal{O}(\bar{R}^2)\,. \tag{57}$$

For the terms involving a background derivative of the regulator we use that

$$\frac{\delta R_k(\bar{\Delta})}{\delta\bar{g}_{\mu\nu}}G_k(\bar{\Delta},\bar{R}=0) = \frac{\delta\bar{\Delta}}{\delta\bar{g}_{\mu\nu}}\frac{\partial R_k(\bar{\Delta})}{\partial\bar{\Delta}}G_k(\bar{\Delta},\bar{R}=0)$$

$$= \frac{\delta}{\delta\bar{g}_{\mu\nu}}\int_0^{\bar{\Delta}}dx'\frac{\partial R_k(x')}{\partial x'}G_k(x',\bar{R}=0)\,, \tag{58}$$

and define the latter integral as

$$F_{\text{RG}}(x) = \int_0^x \mathrm{d}x'\frac{\partial R_k(x')}{\partial x'}G_k(x',\bar{R}=0)\,, \tag{59}$$

where we have restricted ourselves to IR-finite regulators. In straight analogy we manipulate the trace term with $G'$ in (57) and define

$$F_{\mathrm{RG}}^{(1)}(x) = \int_0^x \mathrm{d}x' \, \frac{\partial R_k(x')}{\partial x'} \, G_k^{(0,1)}(x', \bar{R} = 0). \tag{60}$$

As we will see later, these terms only contribute at order $\bar{R}^2$ and are thus not relevant for the present work. This leads as to

$$\mathscr{I}^{\mu\nu} = \frac{1}{4} \, \bar{g}^{\mu\nu} \mathrm{Tr}[R_k G_k] + \frac{1}{2} \mathrm{Tr}\left[\frac{\delta}{\delta \bar{g}_{\mu\nu}} F_{\mathrm{RG}}\right] + \frac{1}{2} \bar{R} \, \mathrm{Tr}\left[\frac{\delta}{\delta \bar{g}_{\mu\nu}} F_{\mathrm{RG}}^{(1)}\right] + \mathcal{O}(\bar{R}^2), \tag{61}$$

where we can pull the $\bar{g}$-derivative out of the trace. In this work we have defined the trace such that we have to there is a factor $\sqrt{\bar{g}}$ involved. Thus we obtain

$$\mathscr{I}^{\mu\nu} = \frac{1}{4} \, \bar{g}^{\mu\nu} \mathrm{Tr}[G_k R_k] + \frac{1}{2} \frac{\delta}{\delta \bar{g}_{\mu\nu}} \mathrm{Tr}[F_{\mathrm{RG}}] - \frac{1}{4} \, \bar{g}^{\mu\nu} \mathrm{Tr}[F_{\mathrm{RG}}]$$

$$+ \frac{1}{2} \bar{R} \frac{\delta}{\delta \bar{g}_{\mu\nu}} \mathrm{Tr}[F_{\mathrm{RG}}^{(1)}] - \frac{1}{4} \, \bar{g}^{\mu\nu} \bar{R} \, \mathrm{Tr}[F_{\mathrm{RG}}^{(1)}] + \mathcal{O}(\bar{R}^2). \tag{62}$$

We now specify these traces to their contributions in order in the background curvature. For this we need to evaluate the background-field derivative. Since $S^4$ is an Einstein manifold, and in particular

$$\bar{R}_{\mu\nu} = \frac{1}{d} \bar{R} \, \bar{g}_{\mu\nu}. \tag{63}$$

Applying a background-field derivative, this gives

$$\frac{\delta \bar{R}}{\delta \bar{g}_{\mu\nu}} = -\frac{1}{d} \bar{R} \, \bar{g}^{\mu\nu} + \bar{g}^{\alpha\beta} \frac{\delta \bar{R}_{\alpha\beta}}{\delta \bar{g}_{\mu\nu}}. \tag{64}$$

We further know that

$$\frac{\delta \bar{R}_{\alpha\beta}}{\delta \bar{g}_{\mu\nu}} = \bar{\Delta}_\rho \frac{\delta \Gamma^\rho_{\alpha\beta}}{\delta \bar{g}_{\mu\nu}} - \bar{\Delta}_\alpha \frac{\delta \Gamma^\rho_{\rho\beta}}{\delta \bar{g}_{\mu\nu}}, \tag{65}$$

are total derivatives and do not contribute. In summary we have

$$\frac{\delta \sqrt{\bar{g}}}{\delta \bar{g}_{\mu\nu}} = \frac{1}{2} \sqrt{\bar{g}} \, \bar{g}^{\mu\nu}, \qquad\qquad \frac{\delta \sqrt{\bar{g}} \bar{R}}{\delta \bar{g}_{\mu\nu}} = \frac{1}{4} \sqrt{\bar{g}} \, \bar{g}^{\mu\nu} \bar{R}. \tag{66}$$

Using this for (62) we obtain

$$I\Big|_{\sqrt{\bar{g}}\text{-terms}} = \frac{1}{4} \mathrm{Tr}[G_k R_k]\Big|_{\sqrt{\bar{g}}\text{-terms}},$$

$$I\Big|_{\sqrt{\bar{g}}\bar{R}\text{-terms}} = \frac{1}{4}\left(\mathrm{Tr}[G_k R_k] - \frac{1}{2} \mathrm{Tr}[F_{\mathrm{RG}}]\right)\Big|_{\sqrt{\bar{g}}\bar{R}\text{-terms}}. \tag{67}$$

Remarkably the $F_{\mathrm{RG}}$-traces do not contribute to the $\sqrt{\bar{g}}$-terms and the $F_{\mathrm{RG}}^{(1)}$-traces drop out completely. The latter would contribute to the $\sqrt{\bar{g}}\bar{R}^2$-order. In summary we compute the traces $\mathrm{Tr}[G_k R_k]$, $\mathrm{Tr}[F_{\mathrm{RG}}]$ and $\mathrm{Tr}[G_k \partial_t R_k]$. The latter we need for the flow of the background couplings, which are also the basis for the level-one flow equations.

With the gauge choice $\beta = 0$ and $\alpha \to 0$, the product of $G_k R_k$ is given by the sum of the contributions of the four modes $h_{\mu\nu}^{\text{tt}}, \xi^\mu, h^{\text{tr}}$, and $\sigma$. They are given by

$$(GR)_{h_{\text{tt}}} = \frac{r_k(x)}{x + r_k(x) - 2\lambda} - \frac{2}{3} \frac{\bar{R}}{k^2} \frac{r_k(x)}{(x + r_k(x) - 2\lambda)^2} + \mathcal{O}(\bar{R}^2),$$

$$(GR)_{\xi} = \frac{r_k(x)}{x + r_k(x)} + \frac{1}{4} \frac{\bar{R}}{k^2} \frac{r_k(x)}{(x + r_k(x))^2} + \mathcal{O}(\bar{R}^2),$$

$$(GR)_{h_{\text{tr}}} = \frac{r_k(x)}{x + r_k(x) - \frac{4}{3}\lambda},$$

$$(GR)_{\sigma} = \frac{r_k(x)}{x + r_k(x)} + \frac{1}{3} \frac{\bar{R}}{k^2} \frac{r_k(x)}{(x + r_k(x))^2} + \mathcal{O}(\bar{R}^2), \tag{68}$$

where $x = \bar{\Delta}/k^2$ is the dimensionless background Laplacian. The vector and scalar ghost modes, $c$ and $\eta$, respectively, are given by

$$(GR)_c = \frac{r_k(x)}{x + r_k(x)} + \frac{1}{4} \frac{\bar{R}}{k^2} \frac{r_k(x)}{(x + r_k(x))^2} + \mathcal{O}(\bar{R}^2),$$

$$(GR)_{\eta} = \frac{r_k(x)}{x + r_k(x)} + \frac{1}{3} \frac{\bar{R}}{k^2} \frac{r_k(x)}{(x + r_k(x))^2} + \mathcal{O}(\bar{R}^2). \tag{69}$$

The generalisations to $G_k \partial_t R_k$ and $G_k \partial_{\bar{\Delta}} R_k$ are straightforward. Throughout this work we use a Litim-type flat cutoff function [110]

$$r_k(x) = (1 - x)\,\theta(1 - x). \tag{70}$$

We start with the evaluation of $\text{Tr}[G_k R_k]$ and write

$$\text{Tr}[G_k R_k] = \frac{1}{16\pi^2}\Big[B_0(\bar{\Delta})Q_2(R_k G_k) + B_2(\bar{\Delta})Q_1(R_k G_k)\Big] + \mathcal{O}(\bar{R}^2), \tag{71}$$

where

$$Q_n[W] = \frac{1}{\Gamma(n)} \int \mathrm{d}x\, x^{n-1}\, W(x),$$

$$B_n = \int \mathrm{d}^d x\, \sqrt{\bar{g}}\, \text{tr}\, b_n, \tag{72}$$

and $\text{tr}\, b_n(\bar{\Delta}_s) \propto \bar{R}^{n/2}$ are the heat kernel coefficients. We parameterise all graviton and ghost modes from (68) and (69) with the function

$$W(x) = \frac{r_k(x)}{[x + r_k(x) + a\,\lambda]^b}, \tag{73}$$

with constants $a$ and $b$. This results in

$$Q_2[W] = \frac{1}{6} \frac{k^{6-2b}}{(1 + a\lambda)^b}, \qquad Q_1[W] = \frac{1}{2} \frac{k^{4-2b}}{(1 + a\lambda)^b}. \tag{74}$$

Furthermore on the sphere we have

$$B_0(\bar{\Delta}_s) = \sqrt{\bar{g}}\, \text{tr}\, b_0, \qquad B_2(\bar{\Delta}_s) = \sqrt{\bar{g}}\, \text{tr}\, b_2, \tag{75}$$

Table 1: Heat kernel coefficients for transverse traceless tensors (TTT), transverse vectors (VT) and scalars (S) on $S^4$.

|            | TTT             | VT            | S             |
|------------|-----------------|---------------|---------------|
| tr $b_0$   | 5               | 3             | 1             |
| tr $b_2$   | $-\frac{5}{6}R$ | $\frac{1}{4}R$ | $\frac{1}{6}R$ |

where the trace coefficients are given in Tab. 1. By specialising the coefficients $a$ and $b$ in the general expression above and evaluating the sum over all spin modes, we find

$$
\text{Tr}[G_k R_k] = \frac{\sqrt{\bar{g}}}{16\pi^2}\left\{\left[\frac{5}{6}\frac{1}{1-2\lambda} + \frac{1}{6}\frac{1}{1-\frac{4}{3}\lambda} - \frac{2}{3}\right]k^4\bar{R}^0\right.
$$
$$
\left. + \left[-\frac{5}{9}\frac{1}{(1-2\lambda)^2} - \frac{5}{12}\frac{1}{1-2\lambda} + \frac{1}{12}\frac{1}{1-\frac{4}{3}\lambda} - \frac{7}{18}\right]k^2\bar{R}^1\right\} + \mathcal{O}(\bar{R}^2). \tag{76}
$$

We turn now to the evaluation of $\text{Tr}[F_{\text{RG}}]$, which is evaluated in similar fashion. We write

$$
\text{Tr}[F_{\text{RG}}] = \frac{1}{16\pi^2} F_{\text{RG}}(-\partial_\tau)\left[B_0(\bar{\Delta}_s)\,\tau^{-2} + B_2(\bar{\Delta}_s)\,\tau^{-1}\right]\Big|_{\tau=0} + \mathcal{O}(\bar{R}^2), \tag{77}
$$

where we used

$$
\text{Tr}f(\bar{\Delta}) = f(-\partial_\tau)\text{Tr}\,e^{-\tau\bar{\Delta}}\Big|_{\tau=0}. \tag{78}
$$

In particular, by utilising the identities

$$
\frac{1}{\tau^2} = \int_0^\infty \mathrm{d}x\, x\, e^{-\tau x}\Big|_{\tau=0},
$$
$$
\frac{1}{\tau} = \int_0^\infty \mathrm{d}x\, e^{-\tau x}\Big|_{\tau=0}, \tag{79}
$$

one evaluates the action of the differential operators and finds

$$
F_{\text{RG}}(-\partial_\tau)\tau^{-2} = \int_0^\infty \mathrm{d}x\, x\, F_{\text{RG}}(x) = -\frac{1}{3}\frac{k^{6-2b}}{(1+a\lambda)^b},
$$
$$
F_{\text{RG}}(-\partial_\tau)\tau^{-1} = \int_0^\infty \mathrm{d}x\, F_{\text{RG}}(x) = -\frac{1}{2}\frac{k^{4-2b}}{(1+a\lambda)^b}, \tag{80}
$$

where the last equality in each line holds for the general function

$$
F_{\text{RG}}(x) = \int_0^x \mathrm{d}y\, \frac{r_k'(y)}{(y + r_k(y) + a\,\lambda\,k^2)^b} = -\frac{x}{(1+a\lambda)^b}, \quad \text{for } x \le k^2, \tag{81}
$$

using the flat Litim-type cutoff as before. Specialising the coefficients $a$ and $b$ for the various spin modes and summing the contributions we get

$$
\text{Tr}[F_{\text{RG}}] = \frac{\sqrt{\bar{g}}}{16\pi^2}\left\{\left[-\frac{5}{3}\frac{1}{1-2\lambda} - \frac{1}{3}\frac{1}{1-\frac{4}{3}\lambda} + \frac{4}{3}\right]k^4\right.
$$
$$
\left. + \left[\frac{5}{12}\frac{1}{1-2\lambda} - \frac{1}{12}\frac{1}{1-\frac{4}{3}\lambda} + \frac{5}{24}\right]k^2\bar{R}^1\right\} + \mathcal{O}(\bar{R}^2). \tag{82}
$$

The evaluated traces (76) and (82) allow us to compute the corrections from the Nielsen identity (67). The result is displayed in (55) where also the contribution from the minimally coupled scalars is shown. The computation of the latter is the same as the scalar graviton modes.

Last we evaluate $\text{Tr}\,[G_k\partial_t R_k]$. which is done in straight analogy to the previous traces. The heat-kernel functionals are now parameterised with

$$\tilde{W}(x) = \frac{\partial_t\,k^2\,r_k(x)}{[x + r_k(x) + a\,\lambda]^b}\,.\tag{83}$$

With $\partial_t\,k^2\,r_k(x) = 2\,k^2\,r_k(x)$ for the flat cutoff, we find

$$Q_2[\tilde{W}] = \frac{k^{6-2b}}{(1+a\lambda)^b}\,,\qquad Q_1[\tilde{W}] = \frac{2\,k^{4-2b}}{(1+a\lambda)^b}\,.\tag{84}$$

where we have suppressed wave-function renormalisations and anomalous dimensions for readability. The result of summing over all spin modes is displayed in (52).

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
