# Peer review of "Effective universality in quantum gravity"

_SciPost Physics, doi:SciPost Phys. 5, 031 (2018)_

## Round 1 · Referee Report · Anonymous · 2018-5-29

Strengths
1) A physical principle, namely the universality of coupling constants due to diffeomorphism invariance, is investigated and effectively verified in asymptotically safe quantum gravity.
2) A highly sophisticated truncation is employed but nevertheless this kind of universality is investigated in the most widely used truncation. The obtained results are highly interesting because it becomes evident that for any quantitative statement very sophisticated truncation schemes are necessary.
3) The manuscript is very well and concisely written.
Weaknesses
1) The expansion around a flat background should be overcome.
2) Only the linear split of the metric is used. Within an exponential split of the metric the background approximation might perform better. I consider this a likely possibility because denominators of the type ``1-2\lambda'' will be absent.
3) Parts of the paper are (maybe unavoidably?) very technical.
Report
The reported results are very interesting, and the paper is well written. Therefore I recommend publication.
However, the manuscript is not reasonably self-contained. It might profit by the additions described below.
Requested changes
The manuscript will profit by adding the following points,
preferentially in Appendix A:
- Add the definition of the Litim regulator. (It might be also appropriate to add a citation to Litim's corresponding paper.)
- Add the York decomposition and the related field redefinitions.
- Explain what is meant by type-I and type-II regulators, and refer in the caption of fig. 7 and the related text in sect. V.B.2 to this explanation.
Author: Manuel Reichert on 2018-07-25 [id 298]
(in reply to Report 1 on 2018-05-29)
We thank the referee for his report and we would like to comment on the pointed out weaknesses.
1) It is indeed an important question to understand whether the observed effective universality persists in a curved background. Such a computation is however technically very challenging and has so far only been achieved in pure gravity in arXiv:1711.09259. The small dependence of the couplings on the background curvature found in that study gives a hint that the observed effective universality might persist. The discovery of effective universality in the present setup makes it a worthwhile endeavour to explore this question more extensively. However, an analysis in a curved background adds the technical difficulties of both computations and is far beyond the scope of this paper.
2) While we agree with the referee that it is a very interesting question to understand the status of effective universality in different parameterisations, we would like to point out that quantum fluctuations also induce a mass parameter for metric fluctuations in the exponential split. In fact, these are the same diagrams as in Fig. 1 of arXiv:1512.01589. They are non-vanishing when evaluated at zero external momentum, i.e., when projected onto a mass parameter. Therefore, the graviton propagator in the exponential split is also of the form $(1+\mu)$, even if $\mu=0$ at the "classical" level, i.e., under an expansion of the Einstein action.
3) The nature of our work makes a less technical presentation very challenging, if the reader should be able to follow our calculations in detail. Therefore some sections are more technical on purpose, as they contain a more detailed presentation. The key results can be understood from the introduction, Sec. II C and the summary.
We thank the referee for the suggested changes and we modified App. A of our manuscript accordingly.
Author: Manuel Reichert on 2018-07-25 [id 299]
(in reply to Report 2 on 2018-06-07)We thank the referee for his report and we would like to comment on the pointed out weaknesses.
1) In our opinion the search for effective universality is well motivated, both from a physical as well as a technical point of view. Our positive results provide a strong indication for the near-perturbative and physical nature of the UV fixed point. We have improved the introduction of our manuscript in order to highlight this more extensively.
2) More extended approximations are certainly a worthwhile extension of our work, and are a natural next step. The referee assesses the present approximation as 'crude', which we actually find surprising.
Let us start our response with a general comment. The systematic error assessment (and hence the assessment of 'crudeness') of any systematic expansion in strongly correlated systems (not only gravity) is a very difficult task, and can only be -partially- done a posteriori, for example with an analysis of the apparent convergence of the expansion. However, we feel that an assessment should take into account the technical status of computations in the given area. In this context we would like to stress the technical sophistication of our study that includes vertices with up to $8\cdot 10^4$ terms as well as full momentum dependence of the two- and three-point functions. These are not only state-of-the art calculations but cutting-edge.
More importantly, the discovery of effective universality in this system hints at the semi-perturbative nature of the fixed point and offers as such an a posteriori justification for our approximation. This exciting possibility of a near-perturbative fixed point now has to be consolidated in further works, and it would point towards the semi-quantitative nature of the present approximation. Finally, extended approximations in many studies in pure gravity already point to the Newton coupling and the cosmological constant as key drivers of the dynamics in the system.
In order to make this important point clear, we have added some further explanations about the near-perturbative nature of the fixed point.
3) We agree that it is an important task to quantify to what extent the modified Slavnov-Taylor identities are satisfied. Our analysis is only based on a comparison of the flows of the couplings. Such computations are highly involved for gravity and the present manuscript is the first to address the question of modified Slavnov-Taylor identities in gravity. Thus this task goes beyond the scope of the manuscript but should certainly be addressed in the future.
4) In Fig. 3 we argued that the error of the applied bilocal projection for the scalar-graviton coupling is of the order 20%. Based on this we call a deviation of this order a "semi-quantitative agreement". We changed our manuscript to make the wording more clear.

---

## Round 1 · Referee Report · Anonymous · 2018-6-7

Strengths
1- the paper reports on what seems to be a great deal of careful algebraic and numerical work
on an approximation to the flow equations for asymptotic safety in quantum gravity, where
the complicated nature of the equations already poses severe challenges.
2- the paper is reasonably well written
Weaknesses
1- the basic effect they are looking for is ill-motivated
2- the approximations are very crude
3- no attempt is made to check whether the modified Slavnov-Taylor identities are actually satisfied
4- a clear statement of what "quantitative agreement" means is missing. From fig. 4 I guess they
mean better than 20%.
Report
The purpose of the paper is to investigate the extent to which two three-point vertices of
the effective action agree numerically for a symmetric configuration of momentum arguments.
This is what they call effective universality and is motivated by the fact that it is true in the classical
action. This is used as a principle to judge the quality of an approximation, and justify a-posteriori
even cruder approximations. However as the authors acknowledge, even if the theory were treated
exactly, such an agreement is no longer expected at the quantum level, these three-point Green
functions being instead related through Slavnov-Taylor identities. Worse still, these identities are
modified by the presence of the infrared cutoff. When the flow equation is investigated in some
severe approximations (retention of only two three-point vertices at only two configurations of
momenta) they find that effective universality is nevertheless satisfied at better than 20%.
Requested changes
1- A more highly motivated research project would quantify the extent to which the modified
Slavnov-Taylor identities are satisfied

---

## Round 2 · List of Changes

- extended motivation in the introduction
- added paragraph about STIs below equation (5)
- included Litim-type regulator and York-decomposition in App. A
- references updated
- other minor changes

---

## Editorial Decision

published